# Layer-Informed Fine-Tuning via Three-Stage Functional Segmentation of LLMs

## Abstract

In recent years, the performance of large language models (LLMs) on reasoning tasks has been remarkable, even surpassing human capabilities on various benchmarks. However, there remains a lack of clear understanding in the academic community regarding how the structure and internal parameters of LLMs progressively solve complex reasoning problems. In this study, we investigate the inference process of LLMs on cross-linguistic materials and propose the hypothesis that LLM layers exhibit a structured division of labor across conceptualization, reasoning, and textualization. Conceptualization layers are crucial for transforming natural language inputs into abstract representations within the LLM, while reasoning layers play a key role in reasoning over these abstract concepts. Finally, the textualization layers convert these abstract representations back into natural language. Based on this hypothesis, we propose a novel approach, LIFT, to achieves efficient and effective finetuning by selectively finetuning only those layers most relevant to a given task's functionality. We then conduct extensive experiments to show that the LIFT method not only accelerates the training process but also significantly improves model performance.

## 1 Introduction

In recent years, large language models (LLMs) (Meta, 2024; Bai et al., 2023; Achiam et al., 2023; Abdin et al., 2024; Guo et al., 2025) have demonstrated outstanding performance in a wide range of tasks and made significant progress in handling complex problems such as reasoning (Wei et al., 2022), mathematics (OpenAI, 2024; LlamaWebsite, 2024), and programming (Guo et al., 2024). Most LLMs utilize a deep, multi-layer transformer architecture (Vaswani et al., 2017), where each layer comprises self-attention mechanisms and feedforward networks that iteratively refine representations and capture increasingly complex linguistic patterns. Research on how LLMs internally comprehend problems, perform reasoning, and ultimately generate correct answers has become a crucial and highly significant topic.

Many studies investigate the mechanistic interpretability of LLMs using methods such as the logit lens (Nostalgebraist, 2020), structural probing (Hewitt & Manning, 2019), and cosine-similarity analyses between hidden states (Timkey & Van Schijndel, 2021). For example, Wendler et al. (2024) found that when generating non-English text, the logit-lens-decoded tokens in the final transformer layers of the LLM initially resemble English before transitioning into the target language in the last few layers. Another study (Li et al., 2024) focuses on the safety of LLMs and finds that certain consecutive intermediate layers may have the ability to detect harmful questions and decide whether to refuse to answer them. In addition, Jin et al. (2024) finds that complex problems may require more layers for comprehension and reasoning, whereas simple problems can be processed and resolved within just a few layers. Parallel to interpretability studies, recent works (Fan et al., 2024; Gromov et al., 2024; Men et al., 2024) have explored the efficiency and redundancy of large language models, particularly in layers closer to the output, enabling efficient pruning strategies.

Inspired by these studies, we investigate the functional division of different layers within an LLM. We propose and provide empirical evidence for the hypothesis that LLM layers exhibit a structured division of labor across conceptualization, reasoning, and textualization. We refer to this hypothesis as the Three-Stage Functional Segmentation of LLMs. According to this hypothesis, an LLM can be divided into three contiguous sections: (1) conceptualization layers, located near the input, (2)

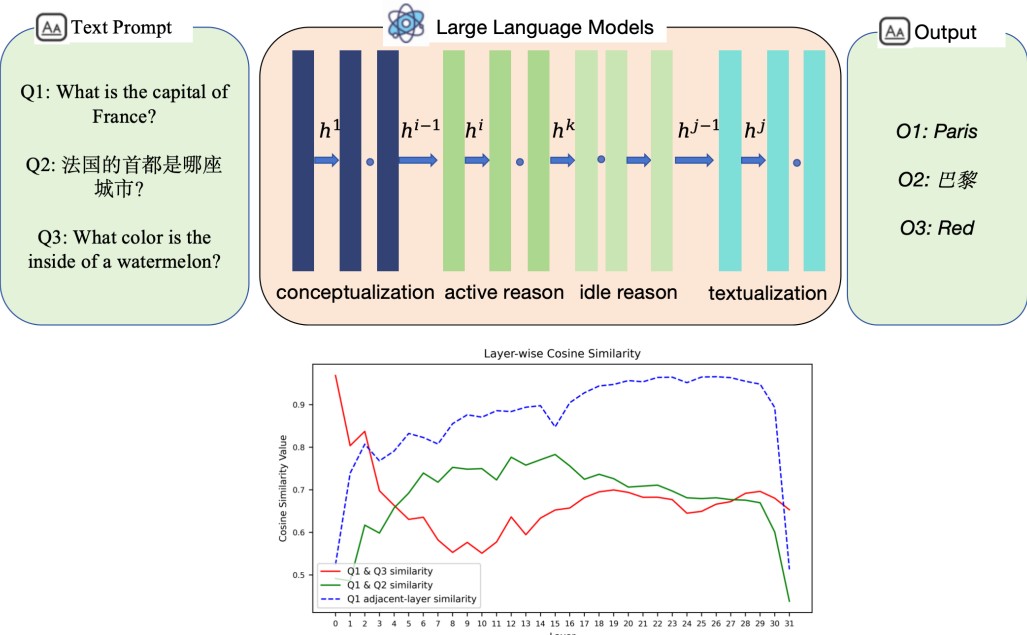

Figure 1: The upper part of the figure illustrates how an LLM processes a simple question under the Three-Stage Functional Segmentation hypothesis. The lower part presents a toy experiment validating this hypothesis by comparing hidden state similarities across conceptualization, reasoning, and textualization layers. We have also added a curve showing the cosine similarity between adjacent layers, with higher values indicating greater redundancy in those layers. In the figure, Q2 and O2 are the Chinese expressions of Q1 and O1, respectively.

reasoning layers, located in the middle, and (3) textualization layers, positioned near the output. The conceptualization layers transform natural language into abstract representations, encoding task-relevant information to facilitate subsequent reasoning. The reasoning layers then process these abstract representations, deriving an intermediate conceptual form that aligns with the intended output. Finally, the textualization layers convert these abstract representations back into natural language, producing the final output. In addition, driven by the observed redundancy (especially in layers closer to the output) in LLM reasoning, we further split the reasoning stage into active and idle reasoning layers. Active reasoning layers carry out the main inference work, whereas idle reasoning layers add little or no new information. Following Jin et al. (2024), we hypothesize that the proportion of idle reasoning layers scales with task complexity, shrinking for harder tasks and growing for easier ones.

To illustrate this hypothesis with a concrete example, consider the input question: "What is the capital of France?" According to the hypothesis, the conceptualization layers first transform the natural language query into an abstract concept, which is articulated as "querying the capital of France." Next, the reasoning layers process this abstract concept to infer the correct answer in its abstract concept form, which in this case corresponds to the abstract concept of "Paris." Finally, the textualization layers convert this abstract concept back into natural language, generating the output "Paris."

The upper part of Figure 1 provides a detailed illustration of how an LLM processes this simple question under the Three-Stage Functional Segmentation hypothesis. The lower part presents a toy experiment designed to empirically support this hypothesis. Since the conceptualization layers map natural language inputs into abstract concepts, two semantically equivalent questions in different languages (Q1 and Q2) should converge to a similar abstract concept of "querying the capital of France." Consequently, their hidden states should become more similar as they pass through the conceptualization layers. In contrast, for two questions in the same language but with different meanings (Q1 and Q3), the similarity of their hidden states should decrease at this stage. At the reasoning layer, where the model derives and refines the abstract concept corresponding to the correct answer, the hidden state similarity between Q1 and Q3 should remain relatively low, while Q1 and Q2 have undergone similar reasoning processes, and should maintain higher similarity. Finally, in

the textualization layers, where the abstract concept is mapped back into natural language, Q1 and Q3 should show an increase in similarity, whereas Q1 and Q2 should diverge as their outputs are rendered in different languages. In addition, we observe that adjacent-layer cosine similarity peaks in the layers near the textualization stage (roughly layers 17–29), indicating that these layers add little novel information and exhibit high redundancy, corresponding to the idle reasoning layers.

Following experiments that support the Three-Stage Functional Segmentation hypothesis, we propose a method for locating the conceptualization, reasoning (further partitioned into active/idle subsets), and textualization layers. Based on the identified approximate positions of these functional layers, we introduce Layer-Informed Fine-Tuning (LIFT), which enables efficient and effective adaptation of LLMs by selectively fine-tuning the most functionally critical layers for a given task—whether they are conceptualization layers, active reasoning layers, or other specialized modules.

Our contributions can be summarized in three key aspects:

- We propose the hypothesis that LLMs can be functionally divided into conceptualization layers, reasoning layers, and textualization layers. Moreover, for each task, we further refined the reasoning layers by partitioning them into active and idle subsets.

- Our experiments provide strong support for our hypothesis and present a method for approximately identifying the ranges of the conceptualization layers, active reasoning layers, idle reasoning layers and textualization layers.

- We propose LIFT, an approach that achieves efficient and effective fine-tuning by selectively fine-tuning only those layers most relevant to a given task's functionality. This method not only reduces fine-tuning time cost but also enhances performance.

## 2 RELATED WORKS

**Internal Structure of LLMs** Research on the mechanistic interpretability of LLMs often utilizes techniques such as the logit lens (Nostalgebraist, 2020), probing (Hewitt & Manning, 2019), and cosine similarity (Timkey & Van Schijndel, 2021) to explore their internal structures. For instance, studies posit that intermediate layers form a "semantic hub" where inputs from diverse languages and modalities are mapped to a shared space (Wu et al., 2024), which is geometrically characterized as a "high-dimensional abstraction phase" critical for initial linguistic processing and downstream task transferability (Cheng et al., 2024). The effectiveness of these mid-depth representations is demonstrated by their superior performance on downstream tasks relative to final-layer outputs (Skean et al., 2025), with models also dynamically engaging deeper layers to address more complex problems (Jin et al., 2024). A common finding within these frameworks is the phenomenon of a latent operational language. English-centric models tend to "think" in English within their abstract middle layers, only translating concepts into the target language in the final layers (Wendler et al., 2024; Zhao et al., 2024; Zhong et al., 2024). This behavior extends to non-English-centric models, which may develop dual latent languages corresponding to their training data, activating the language most similar to the target output during processing (Zhong et al., 2024). Further evidence for layer specialization comes from diverse analytical perspectives. Adopting a neuroscientific approach, AlKhamissi et al. (2024) employed functional localizers to identify and causally validate "language-selective units" concentrated in specific layers that are critical for linguistic tasks. These studies provide a valuable "map" of the functional properties contained within different layers.

Concurrently, research on model robustness has found that early and final layers are highly sensitive to structural interventions like deletion or swapping, whereas middle layers are remarkably resilient (Sun et al., 2025; Lad et al., 2024). This resilience is particularly pronounced in the later-middle layers, which have been shown to be more redundant and less impactful on performance when pruned compared to earlier layers (Gromov et al., 2024; Men et al., 2024), further supporting their distinct functional role.

While these studies provide a valuable "map" of the functional properties within LLM layers, our work provides a "recipe" for using them. We shift the focus from a static analysis of what is in each layer to the dynamic transitions between these functional stages. This approach allows us to develop targeted and parameter-efficient fine-tuning strategies that improve performance by focusing updates only on functionally-relevant blocks.

**Parameter-Efficient Fine-Tuning.** The conventional full-parameter fine-tuning paradigm often incurs high computational costs when fine-tuning LLMs. To address this, parameter-efficient fine-tuning (PEFT) has emerged as a promising alternative. Key PEFT approaches include Adapter Tuning (Houlsby et al., 2019; Pfeiffer et al., 2020; Karimi Mahabadi et al., 2021), Prompt Tuning (Li et al., 2023), and Low-Rank Adaptation (LoRA) (Hu et al., 2021). Adapter Tuning introduces trainable adapter modules between all model layers, Prompt Tuning incorporates tunable prefix tokens into inputs.The original LoRA (Hu et al., 2021) decomposes the weight updates for various parameter matrices into low-rank adaptation matrices. More advanced variants like AdaLoRA (Zhang et al., 2023) improve upon this by dynamically allocating the parameter budget to more important weight matrices based on their sensitivity during training. These methods only need to fine-tune a small subset of parameters, typically less than 1% of the original, and can significantly reduce computational and memory overhead.

Compared with the above methods, our LIFT focuses on fine-tuning specific layers in the middle, which follows a distinct trajectory from the above methods and can be seamlessly integrated with them. For instance, in this paper, we incorporate LoRA to achieve both high performance and accelerated fine-tuning.

## 3 PRELIMINARY

### 3.1 LARGE LANGUAGE MODELS

Large Language Models, such as Llama-3 (Meta, 2024), are built on a transformer-based architecture composed of multiple stacked layers. Each layer consists of a multi-head self-attention mechanism that captures contextual dependencies, a feedforward network for further processing, and residual connections with layer normalization to stabilize training. In this framework, hidden states represent the evolving internal representations of tokens, which are progressively enriched layer by layer. These states are updated iteratively by adding residuals, where the residual is computed as a function of the hidden states of all preceding tokens. In this paper, we employ four LLMs from different model families: Llama-3-8B-Instruct (Meta, 2024), Phi-3-mini-4k-instruct (Abdin et al., 2024), and Qwen2-7B-Instruct (Yang et al., 2024). These four LLMs were developed by different companies, each with distinct internal architectures and training methodologies, making them highly representative of diverse model designs.

### 3.2 DATASETS

To train and evaluate the reasoning capabilities of LLMs, we select three representative logical reasoning datasets: ProofWriter (Tafjord et al., 2020), FOLIO (Han et al., 2022) and the LogicalDeduction dataset from BigBench (Srivastava et al., 2022). Our data processing method follows a similar approach to that proposed in Pan et al. (2023). These datasets ensures a comprehensive assessment of the model's ability to perform various forms of logical reasoning.

- **ProofWriter** (Tafjord et al., 2020) is a widely used dataset for deductive logical reasoning. In our study, we utilize the open-world assumption subset. This dataset is categorized into five classes based on the depth of reasoning required. To ensure a rigorous evaluation, we select the most challenging subset, depth-5.
- **FOLIO** (Han et al., 2022) is a challenging dataset for first-order logic reasoning, featuring expert-written problems that require complex logical deductions. The questions are presented in natural language and are closely aligned with real-world knowledge.
- **LogicalDeduction** is a logical reasoning task from the BigBench benchmark (Srivastava et al., 2022), focusing on constraint satisfaction problems. The task primarily involves deducing the order of a sequence of objects given a minimal set of conditions.

In addition to the reasoning dataset, we also conducted experiments on GSM8K, a mathematical dataset.

- **GSM8K** (Cobbe et al., 2021): This dataset contains 8,500 grade school-level math problems that cover a wide range of topics, including arithmetic, algebra, and geometry.

# 4 THREE-STAGE FUNCTIONAL SEGMENTATION OF LLMS

In this section, we present our hypothesis, the Three-Stage Functional Segmentation of LLMs, and provide rigorous and comprehensive experiments to support the hypothesis.

According to this hypothesis, an LLM can be functionally divided into three contiguous stages:

1. **Conceptualization layers** – located near the input, responsible for transforming natural language into abstract representations.

2. **Reasoning layers** – situated in the middle, where task-relevant logical processing occurs. These can be further divided into **active reasoning layers** and **idle reasoning layers**.

3. **Textualization layers** – positioned near the output, where abstract representations are mapped back into natural language.

The conceptualization layers encode linguistic inputs into abstract, task-relevant representations. This initial stage marks a functional transition from processing surface-level language to forming the conceptual foundation required for logical processing.

The reasoning layers then manipulate these abstract representations to perform logical inference. Operating on task-specific logic rather than surface linguistics, we hypothesize that hidden states in this stage will exhibit high similarity for the same task across different languages, yet low similarity for different tasks within the same language. Furthermore, these layers can be subdivided into active reasoning layers, which execute core inference, and idle reasoning layers, which are functionally redundant.

Finally, the textualization layers translate the inferred abstract concepts back into natural language. This final stage reverses the similarity pattern observed during reasoning: representations for different tasks converge as they are mapped to a common linguistic form, while those for the same task in different languages diverge into their respective linguistic outputs.

To investigate the functional segmentation of layers within the LLM, we design the following experiments during the inference process. Suppose a large language model has $K$ hidden layers and a task-consistent dataset $\mathcal{D}_{\text{raw}} = \{d_i\}_{i=1}^N$. Without loss of generality, assume that this dataset is an English dataset. We then define its corresponding datasets in other natural languages as $\mathcal{D}_{\text{Non-English}} = \{d_i'\}_{i=1}^N$. The Non-English datasets may include any non-English language supported by the LLM, such as Chinese, French, Spanish, or Hindi. The identical index $i$ in these datasets indicates the same semantic content, differing only in language.

When using these two datasets as the input for inference in the LLM, we obtain the hidden state sets from the last position of each layer in the first autoregressive process, denoted as $S(D_{\text{raw}})$ and $S(D_{\text{Non-English}})$. The specific form of these vector sets is as follows:

$$S(D_{\text{raw}}) = \left\{ [h_{(d_i)}^1, h_{(d_i)}^2, \ldots, h_{(d_i)}^K] \right\}_{i=1}^N, \qquad S(D_{\text{Non-English}}) = \left\{ [h_{(d_i')}^1, h_{(d_i')}^2, \ldots, h_{(d_i')}^K] \right\}_{i=1}^N$$

where $h_{(d_i)}^k$ represents the hidden state at the last position in layer $k$ during the first autoregressive process.

## 4.1 COSINE SIMILARITY BETWEEN HIDDEN STATES IN THE SAME LAYER

First, we compute the cosine similarity for the following pairs. These results quantify the similarity between hidden representations across different layers, offering insights into how information is processed and transformed within the model.

1. For pairs of the same question in different languages, we randomly select an index $i$ from 1 to $N$ and compute the cosine similarity between different language versions of the same question $i$, denoted as $g_i^k = \cos(h_{(d_i)}^k, h_{(d_i')}^k)$ for $1 \leq k \leq K$.

2. For pairs of different questions within the English dataset, we randomly select two indices $i \neq j$ from 1 to $N$ and compute the cosine similarity between different question $i$ and $j$ in English, denoted as $r_{i,j}^k = \cos(h_{(d_i)}^k, h_{(d_j)}^k)$ for $1 \leq k \leq K$.

3. For pairs of different questions within the Non-English dataset, we randomly select two indices $i \neq j$ from 1 to $N$ and compute the cosine similarity between different question $i$ and $j$, denoted as $b_{i,j}^k = \cos(h_{(d_i')}^k, h_{(d_j')}^k)$ for $1 \leq k \leq K$.

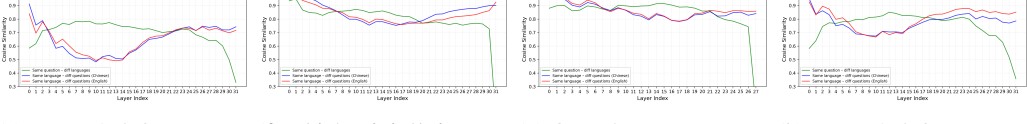

(a) LLaMA-3-8B-Instruct on GSM8K    (b) Phi-3-mini-4k-instruct on GSM8K    (c) Qwen2-7B-Instruct on LogicalDeduction    (d) LLaMA-3-8B-Instruct on LogicalDeduction

(e) LLaMA-3-8B-Instruct on FOLIO    (f) Phi-3-mini-4k-instruct on FOLIO    (g) Qwen2-7B-Instruct on ProofWriter    (h) LLaMA-3-8B-Instruct on ProofWriter

Figure 2: Cosine similarity between Hidden States in the Same Layer

To substantiate our hypothesis, we conduct extensive experiments across multiple models, languages, and datasets. Specifically, we evaluate four widely used models from different model families: Llama-3-8B-Instruct (Meta, 2024), Phi-3-mini-4k-instruct (Abdin et al., 2024) and Qwen2-7B-Instruct (Yang et al., 2024). Furthermore, we assess model behavior across five diverse languages (English, Chinese, French, Spanish, and Hindi) and three reasoning-focused datasets: ProofWriter (Tafjord et al., 2020), FOLIO (Han et al., 2022) and LogicalDeduction (Srivastava et al., 2022). By systematically analyzing model behavior across these varied conditions, we establish robust empirical evidence for our proposed hypothesis. The results are shown in Figure 2, demonstrating a high degree of consistency across models, languages, and datasets, indicating their robustness and broad applicability. To mitigate the impact of experimental randomness, we randomly repeat the experiments 1000 times and take the average of $g_i^k, r_{i,j}^k, b_{i,j}^k$ for the random $i, j$'s.

As shown in Figure 2, the green line is the cosine similarity between the first kind of pairs ($g_i^k$'s for different layer $k$), the red line is the cosine similarity between the second kind of pairs ($r_{i,j}^k$'s for different layer $k$), and the blue line is the cosine similarity between the third kind of pairs ($b_{i,j}^k$'s for different layer $k$).

Our analysis reveals a fundamental divergence between the similarity trends of task-relevant and task-irrelevant pairs. The task-relevant similarity (green line) follows a clear trajectory: it rises to form a distinct plateau throughout the middle layers and subsequently declines in the final layers. Conversely, the task-irrelevant similarity (red and blue lines) exhibits an opposite, U-shaped pattern of decreasing and then increasing. We note that the initial values can be variable, a phenomenon we hypothesize is an artifact of the early hidden states' sensitivity to tokenization differences. Our core claim, however, is based on these starkly opposing trends, which are consistent across languages and strongly substantiate our three-stage hypothesis.

We also investigate the internal representations of these transformer-based models by visualizing hidden states obtained from different language versions of the same dataset, by employing three widely used dimensionality reduction techniques: Principal component analysis (PCA) (Hotelling, 1933), t-distributed stochastic neighboring embedding (t-SNE) (Van der Maaten & Hinton, 2008), and uniform manifold approximation and projection (UMAP) (McInnes et al., 2018). The visualization results are consistent with the similarity results in Figure 2, and we defer them to Appendix.

## 4.2 COSINE SIMILARITY BETWEEN HIDDEN STATE IN ADJACENT LAYERS

To achieve a more refined segmentation of the reasoning layers, we adopt the adjacent-layer cosine similarity analysis similar to Gromov et al. (2024) and Men et al. (2024). For each dataset in subsection 3.2, we compute the cosine similarity between every pair of successive layers. Due to

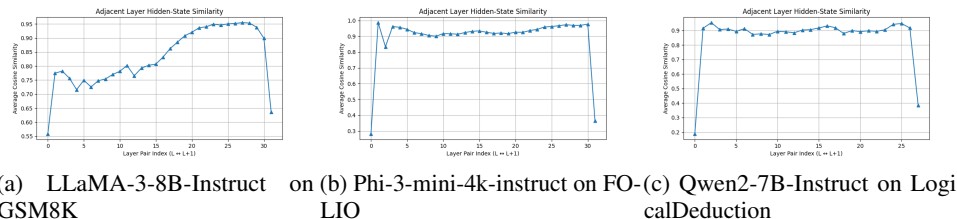

(a) LLaMA-3-8B-Instruct on GSM8K

(b) Phi-3-mini-4k-instruct on FO-LIO

(c) Qwen2-7B-Instruct on Logi-calDeduction

Figure 3: Cosine Similarity between Hidden Stated in Adjacent Layers

the residual architecture of LLMs, a very high similarity between adjacent layers indicates that the latter layer contributes little new information and is therefore highly redundant. Consistent with the findings of Fan et al. (2024), Gromov et al. (2024), and Men et al. (2024), from Figure 3, we observe that reasoning layers near the textualization stage exhibit similarities exceeding 95% (e.g., layers 23-28 in Figure 3a and layers 24-30 in Figure 3b). We classify these high-similarity layers as idle reasoning layers, while the remaining reasoning layers with lower adjacent-layer similarity are designated as active reasoning layers.

### 4.3 SUMMARY OF EXPERIMENTS

In summary, our extensive analysis across multiple models, languages, and datasets provides compelling evidence for the Three-Stage Functional Segmentation hypothesis. The observed dynamics of representation similarity systematically align with our proposed functions: an initial shift from linguistic to abstract conceptual encoding (conceptualization), followed by a stage of stable, language-agnostic processing (reasoning), and a final reversion to linguistic formatting for output (textualization). This pattern not only validates the functional roles of each stage but also reveals the existence of idle reasoning layers where information processing plateaus.

Moreover, this segmentation suggests that reasoning layers are critical for logical inference, as they process task-specific abstract concepts before mapping them to output representations. We may improve reasoning efficiency and generalization by focusing on reasoning layers in LLM. Due to the inherent redundancy of large language models, particularly focusing on their active reasoning layers. In addition, this segmentation further implies that the conceptualization layers are critical for an LLM's language understanding, as they mediate the transformation of natural language into abstract representations. By concentrating on these conceptualization layers, we may also be able to enhance the language comprehension capabilities of large language models.

## 5 LIFT: LAYER-INFORMED FINE TUNING

Building upon the hypothesis of functional segmentation proposed in the previous section, we introduce a fine-tuning method aimed at achieving higher efficiency and improved performance: Layer-Informed Fine Tuning (LIFT). This method *selectively fine-tuning only those layers most relevant to a given task's functionality.* In this paper, we focus exclusively on applying LIFT to the conceptualization layers and the reasoning layers. We denote LIFT[a, b] to represent the fine-tuning process restricted to layers from $a$ to $b$ (excluding layer $b$).

### 5.1 LAYER PARTITION

To implement LIFT, it is essential to first identify the location of functional layer segmentation. We employed a straightforward approach to delineate the layer boundaries. To determine the boundaries between the conceptualization layers and the reasoning layers, we employ the cosine similarity between different language versions of the same question $g^k$ (as described in Section 4), represented by the green line in Figure 2. As previously mentioned in Section 4.1, the green line reflects task-relevant information, which gradually increases within the conceptualization layers and remains relatively stable in the reasoning layers. Therefore, the inflection point during its growth phase can be considered a marker for the functional transition from conceptualization to reasoning. For

this discrete case, we apply the discrete second derivative method to identify the inflection point (Weisstein). First, we compute the discrete second derivative of the layer-wise similarity $g^k$ by

$$\Delta^2 g^k \;=\; g^{k+1} \;-\; 2\,g^k \;+\; g^{k-1}.$$

Second, we identify the first index $k^*$ where $\Delta^2 g^k$ changes sign, i.e., where $\Delta^2 g^{k-1}\,\Delta^2 g^k < 0$ as the inflection point marking the transition from conceptualization to reasoning.

Next, to partition the reasoning block into active layers and idle layers, we utilize the cosine similarity between hidden states in adjacent layers described in Section 4.2. A higher cosine similarity between adjacent layers indicates greater redundancy. Therefore, we set a threshold value of $\beta$ and classify consecutive layers exceeding this threshold as idle reasoning layers, and simply select $\beta = 0.94$. This value was chosen empirically, as our observations confirmed that $\beta = 0.94$ effectively distinguishes the start of the flat, high-similarity region characteristic of idle reasoning layers. During our experiments, we observed that for certain datasets, the adjacent cosine similarity of Qwen2-7B-Instruct did not surpass the threshold $\beta$. In such cases, we conclude that the dataset does not contain any idle reasoning layers. Based on these methods, we can obtain the specific functional layer segmentation for different tasks across various models. Due to space limitations, the detailed segmentation results are provided in the Appendix.

## 5.2 FINETUNING SETTING

To accelerate the finetuning process, we opted to use the popular LoRA framework. LoRA efficiently fine-tunes large models by introducing low-rank updates, thereby reducing memory usage, speeding up training, and improving generalization with minimal parameter changes. For the LIFT method, we apply LoRA layers exclusively to the reasoning layers of the model. In the case of fully finetuning all transformer layers (FullFT), we add LoRA to all the transformer layers.

One important question in the LIFT method is to choose which layers to finetune. For reasoning datasets, given their relatively simple linguistic structure but complex reasoning relationships, our LIFT algorithm is expected to prioritize active reasoning layers for training. However, for mathematics datasets like GSM8K, determining which layers to finetune requires more consideration. According to Srivatsa & Kochmar (2024), the attributes of GSM8K mathematical problems can be primarily divided into linguistic features and mathematical features. Their study explored the correlation between these attributes and the accuracy of LLM responses, revealing that the top-ranked feature in terms of importance belongs to linguistic features. This finding suggests that the major factor influencing LLM accuracy in solving GSM8K is linguistic complexity, hinting that enhancing the conceptualization layers might be more impactful for improving LLMs' ability to solve GSM8K.

To further investigate whether solving GSM8K relies more on linguistic capabilities or mathematical reasoning, we designed the following experiment. To control for variables, our experiment is based on the templates provided by Mirzadeh et al. (2024), which allow us to modify only the numerical components of GSM8K or increase linguistic complexity by adding irrelevant statements. We conducted experiments on LLaMA-3-8B-Instruct (Meta, 2024) and Phi-3-mini-4k-instruct(Abdin et al., 2024), first selecting problems that each model answered correctly with over 90% accuracy (GSM8K-easy), indicating that the LLM had fully mastered the solution. Using the templates of these problems, we generated two distinct datasets: GSM8K-hard-number, where simple numbers were replaced with more complex ones, and GSM8K-hard-language, where irrelevant sentences were added to the original problems. We fine-tuned both models on the GSM8K dataset and subsequently evaluated them on GSM8K-hard-number and GSM8K-hard-language. This approach allows us to determine which of the linguistic features or mathematical features is primarily enhanced by the fine-tuning process.

Table 1 presents the performance of the fine-tuned model on GSM8K-hard-number and GSM8K-hard-language. We observe that fine-tuning has limited effectiveness in enhancing the model's capability on GSM8K-hard-number but significantly improves its performance on GSM8K-hard-language.These findings suggest that improving language capabilities is more crucial. Therefore, we prioritize LIFT optimization specifically for the conceptualization layers when dealing with GSM8K.

## 5.3 EXPERIMENTAL RESULTS

Table 2 presents the performance of the original model, full transformer layer fine-tuning by LoRA, and LIFT by LoRA. We observe that the LIFT approach demonstrates a noticeable improvement in

| | Llama-3-8B-Instruct | | Phi-3-mini-4k-instruct | |
|---|---|---|---|---|
| GSM8K-hard-number | Base model | FullFT | Base model | FullFT |
| Accuracy | 75% | 45% | 52.75% | 45.5% |
| GSM8K-hard-language | Base model | FullFT | Base model | FullFT |
| Accuracy | 56.67% | 63.33% | 72.25% | 71% |

Table 1: Performance Comparison of GSM8K-hard-number and GSM8K-hard-language

| | Llama-3-8B-Instruct | | | Phi-3-mini-4k-instruct | | | Qwen2-7B-Instruct | | |
|---|---|---|---|---|---|---|---|---|---|
| ProofWriter | Base model | FullFT | LIFT[2,20] | Base model | FullFT | LIFT[2,26] | Base model | FullFT | LIFT[3,26] |
| Accuracy | 45.67% | 56.12% (±2.02%) | 57.67% (±0.36%) | 58.33% | 62.75% (±2.30%) | 64.88% (±0.72%) | 48.00% | 53.17% (±3.10%) | 53.46% (±2.36%) |
| Training Time(s) | - | 1383 | 1215 | - | 1107 | 1023 | - | 1288 | 1180 |
| FOLIO | Base model | FullFT | LIFT[2,20] | Base model | FullFT | LIFT[2,23] | Base model | FullFT | LIFT[3,26] |
| Accuracy | 55.39% | 52.08% (±1.67%) | 58.21% (±2.35%) | 48.53% | 62.50% (±2.42%) | 64.09% (±1.09%) | 34.31% | 44.98% (±0.74%) | 44.73% (±1.98%) |
| Training Time(s) | - | 332 | 290 | - | 249 | 225 | - | 310 | 284 |
| LogicalDeduction | Base model | FullFT | LIFT[2,22] | Base model | FullFT | LIFT[2,24] | Base model | FullFT | LIFT[3,24] |
| Accuracy | 46.33% | 53.25% (±1.89%) | 53.58% (±2.47%) | 46.67% | 61.25% (±1.00%) | 61.75% (±2.49%) | 50.33% | 53.83% (±1.23%) | 57.08% (±3.29%) |
| Training Time(s) | - | 516 | 460 | - | 390 | 357 | - | 477 | 428 |
| GSM8K | Base model | FullFT | LIFT[0,2] | Base model | FullFT | LIFT[0,2] | Base model | FullFT | LIFT[0,3] |
| Accuracy | 49.96% | 67.76% (±0.57%) | 73.96% (±0.98%) | 79.23% | 77.14% (±0.34%) | 78.34% (±0.73%) | 78.85% | 74.53% (±0.34%) | 75.34% (±0.66%) |
| Training Time(s) | - | 3347 | 2667 | - | 2817 | 2325 | - | 3500 | 2785 |

Table 2: Performance and Training Time Comparison of LoRA Fine-Tuning Methods over 4 Seeds

performance compared to FullFT, while achieving an average reduction of 12.2% in time cost. This efficiency gain stems from LIFT updating drastically fewer trainable parameters (e.g., only 10.5M vs. 167.8M for Llama-3 on GSM8K), which substantially reduces the memory footprint. These experimental results indicate that by selectively fine-tuning only those layers most relevant to a given task's functionality, LIFT can simultaneously enhance model accuracy, reduce training time, and lower memory requirements.

To further validate the effectiveness of our proposed LIFT algorithm, we conducted ablation experiments by fine-tuning and testing layers outside of the LIFT-selected regions. The detailed results are provided in the Appendix. From these experiments, it is evident that our LIFT algorithm significantly outperforms the alternatives. The fine-tuning strategy, which focuses explicitly on functionally relevant layers, demonstrates remarkable improvements in performance. This underscores the importance of targeting specific functional layers in LLMs for task-specific optimization.

## 6 CONCLUSION AND LIMITATION

In this work, we introduced a novel hypothesis that LLMs can be functionally segmented into three distinct layers: conceptualization, reasoning (further partitioned into active/idle subsets), and textualization. Our experiments validated this hypothesis and demonstrated a method for approximately identifying the ranges of these segmented layers. To leverage this functional segmentation, we proposed LIFT, a selective fine-tuning approach that optimally targets the most task-relevant layers. LIFT not only enhances fine-tuning efficiency by reducing time costs but also improves task-specific performance. These findings mark a significant step forward in the development of adaptive, task-focused language models.

On the other hand, this paper examines only four relatively small-scale open-source models with fewer layers. Although these models have a certain degree of representativeness due to belonging to different model families, further research is required to determine whether the proposed method is applicable to larger-scale models, models with different architectures, and even closed-source models. Furthermore, the effectiveness of fine-tuning textualization layers was not explored in this study, primarily due to the lack of suitable benchmarks for automatically evaluating tasks related to output style, which we leave as a direction for future work.

ETHICS STATEMENT

This research adheres to the ICLR Code of Ethics. Our work centers on the mechanistic interpretability of large language models and the development of more efficient fine-tuning methodologies. The datasets utilized in our experiments—ProofWriter, FOLIO, LogicalDeduction, and GSM8K—are well-established, publicly available benchmarks for evaluating logical and mathematical reasoning. These datasets do not contain any personally identifiable or sensitive information. The multilingual datasets were generated by translating these public benchmarks, and we have made them available for public review. The proposed LIFT method aims to improve computational efficiency, which can contribute positively by reducing the resources required for model adaptation. We do not foresee any direct negative societal impacts or ethical concerns arising from our study.

REPRODUCIBILITY STATEMENT

We have made every effort to ensure the reproducibility of our research. All models used in this study (Llama-3-8B-Instruct, Phi-3-mini-4k-instruct, Qwen2-7B-Instruct) are publicly available, as described in Section 3.1. The datasets are standard benchmarks, detailed in Section 3.2, with further examples provided in Appendix B. Our multilingual datasets and the code used for our cosine similarity analysis are available in the supplementary materials, as noted in Appendix B.2. The methodology for identifying functional layer boundaries is detailed in Section 5.1 and Appendix E. Furthermore, Appendix F provides a comprehensive description of our experimental environment, including hardware specifications, software versions (PyTorch, CUDA), and the complete set of hyperparameters used for LoRA fine-tuning. The supplementary materials also contain the necessary code to replicate our main experiments and generate the figures presented in the paper.

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

# A  LLM STRUCTURE

| Model | Layers | Model | Layers |
|-------|--------|-------|--------|
| Llama-3-8B-Instruct | 32 | Qwen2-7B-Instruct | 28 |
| Phi-3-mini-4k-instruct | 32 | | |

Table 3: Number of layers in each LLM

# B  DATASETS

## B.1  EXAMPLES OF EACH DATASET

### B.1.1  PROOFWRITER

```
"instruction": "The cow is blue. The cow is round. The cow likes the
    lion. The cow visits the tiger. The lion is cold. The lion is nice
    . The lion likes the squirrel. The squirrel is round. The squirrel
     sees the lion. The squirrel visits the cow. The tiger likes the
    cow. The tiger likes the squirrel. If something is cold then it
    visits the tiger. If something visits the tiger then it is nice.
    If something sees the tiger and it is young then it is blue. If
    something is nice then it sees the tiger. If something likes the
    squirrel and it likes the cow then it visits the tiger. If
    something is nice and it sees the tiger then it is young. If the
    cow is cold and the cow visits the lion then the lion sees the
    squirrel.Based on the above information, is the following
    statement true, false, or unknown? The tiger is not young.options:
     ['A) True', 'B) False', 'C) Unknown']",

"input": "",

"output": "B"
```

### B.1.2  FOLIO

```
"instruction": "All people who regularly drink coffee are dependent on
     caffeine. People either regularly drink coffee or joke about
    being addicted to caffeine. No one who jokes about being addicted
    to caffeine is unaware that caffeine is a drug. Rina is either a
    student and unaware that caffeine is a drug, or neither a student
    nor unaware that caffeine is a drug. If Rina is not a person
    dependent on caffeine and a student, then Rina is either a person
    dependent on caffeine and a student, or neither a person dependent
     on caffeine nor a student.Based on the above information, is the
    following statement true, false, or uncertain? Rina is a person
    who jokes about being addicted to caffeine or unaware that
    caffeine is a drug.options: ['A) True', 'B) False', 'C) Uncertain
    ']",

"input": "",

"output": "A"
```

### B.1.3  LOGICALDEDUCTION

```
"instruction": "The following paragraphs each describe a set of five
    objects arranged in a fixed order. The statements are logically
    consistent within each paragraph.On a branch, there are five birds
    : a quail, an owl, a raven, a falcon, and a robin. The owl is the
    leftmost. The robin is to the left of the raven. The quail is the
    rightmost. The raven is the third from the left.Which of the
    following is true?options: ['A) The quail is the rightmost.', 'B)
    The owl is the rightmost.', 'C) The raven is the rightmost.', 'D)
    The falcon is the rightmost.', 'E) The robin is the rightmost.']",

"input": "",

"output": "A"
```

### B.1.4 GSM8K

```
"instruction": "Natalia sold clips to 48 of her friends in April, and
    then she sold half as many clips in May. How many clips did
    Natalia sell altogether in April and May?",
"input": "",
"output": "Natalia sold 48/2 = <<48/2=24>>24 clips in May.\nNatalia
    sold 48+24 = <<48+24=72>>72 clips altogether in April and May.\n
    #### 72"
```

### B.1.5 GSM8K-EASY

```
"input": "When Sophie watches her nephew, she gets out a variety of
    toys for him. The bag of building blocks has 18 blocks in it. The
    bin of stuffed animals has 25 stuffed animals inside. The tower of
     stacking rings has 3 multicolored rings on it. Sophie recently
    bought a tube of bouncy balls, bringing her total number of toys
    for her nephew up to 88. How many bouncy balls came in the tube?",
"output": "Let T be the number of bouncy balls in the tube.\nAfter
    buying the tube of balls, Sophie has 18.0 + 25.0 + 3.0 + T = 46 +
    T = 88.0 toys for her nephew.\nThus, T = 88.0 - 46 =
    <<88.0-46=42.0>>42.0 bouncy balls came in the tube.\n#### 42.0",
```

### B.1.6 GSM8K-HARD-NUMBER

```
"input": "When Sophie watches her nephew, she gets out a variety of
    toys for him. The bag of building blocks has 181 blocks in it. The
     bin of stuffed animals has 193 stuffed animals inside. The tower
    of stacking rings has 279 multicolored rings on it. Sophie
    recently bought a tube of bouncy balls, bringing her total number
    of toys for her nephew up to 986. How many bouncy balls came in
    the tube?",
"output": "Let T be the number of bouncy balls in the tube.\nAfter
    buying the tube of balls, Sophie has 181.0 + 193.0 + 279.0 + T =
    653 + T = 986.0 toys for her nephew.\nThus, T = 986.0 - 653 =
    <<986.0-653=333.0>>333.0 bouncy balls came in the tube.\n####
    333.0",
```

### B.1.7 GSM8K-HARD-LANGUAGE

```
"input": When Sophie watches her nephew, she gets out a variety of
    toys for him, ensuring he has plenty of options to choose from.
    The bag of building blocks has 18 blocks in it, although some are
    slightly worn from previous play sessions. The bin of stuffed
    animals has 25 stuffed animals inside, each representing different
     characters from his favorite storybook. The tower of stacking
```

```
            rings has 3 multicolored rings on it, which he enjoys stacking in
            specific color patterns. Sophie recently bought a tube of bouncy
            balls, adding even more excitement to playtime, bringing her total
             number of toys for her nephew up to 88. It is worth noting that
            she also considered purchasing a set of miniature cars, which were
             on sale, but decided against it due to limited space. How many
            bouncy balls came in the tube?
      "output": Let T be the number of bouncy balls in the tube.\nAfter
            buying the tube of balls, Sophie has 18.0 + 25.0 + 3.0 + T = 46 +
            T = 88.0 toys for her nephew.\nThus, T = 88.0 - 46 =
            <<88.0-46=42.0>>42.0 bouncy balls came in the tube.\n#### 42.0
```

### B.2 MULTILINGUAL DATASET AND AVAILABILITY

To facilitate our multilingual analysis, the original English datasets were translated using the GPT-4 language model. This process generated versions of the datasets in four additional languages: Chinese, French, Spanish, and Hindi. All translated data used in this study is publicly available for review in the supplementary material, located at the following path: Codes/Cosine Similarity/

## C DETAILS OF COSINE SIMILARITIES OF HIDDEN STATES

To make the Cosine Similarity more pronounced, we adjusted the order of the questions in the dataset, presenting the options first followed by the question statement. We present here the complete set of cosine similarities of hidden states from Section 4.

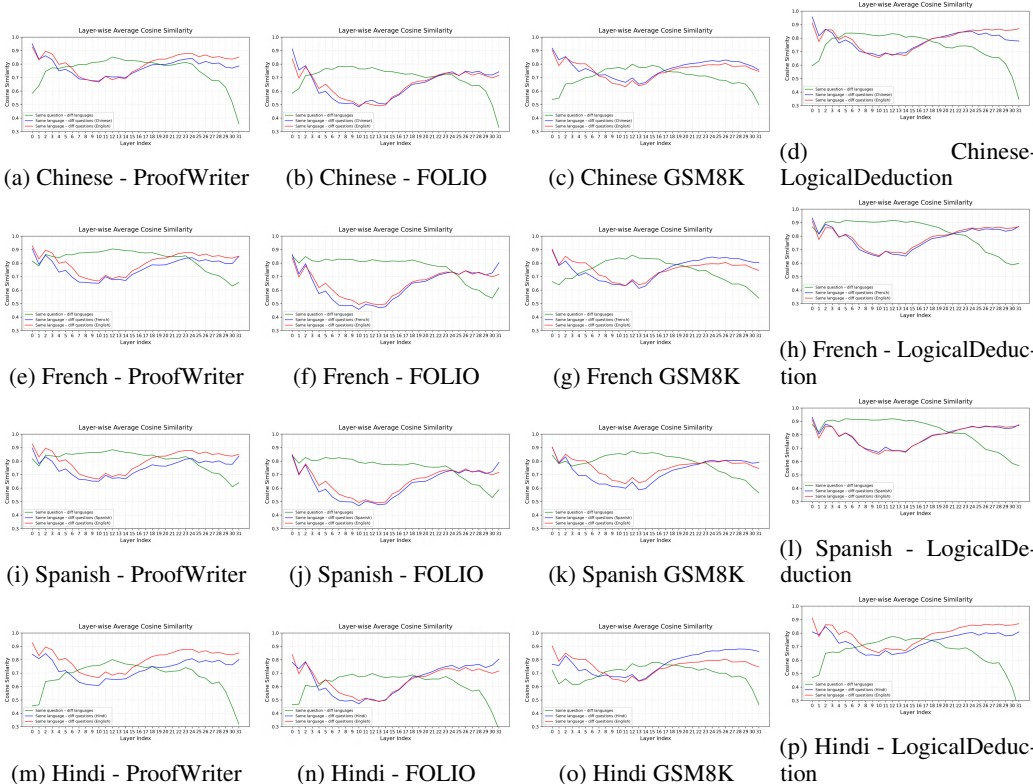

(a) Chinese - ProofWriter    (b) Chinese - FOLIO    (c) Chinese GSM8K    (d) Chinese-LogicalDeduction

(e) French - ProofWriter    (f) French - FOLIO    (g) French GSM8K    (h) French - LogicalDeduction

(i) Spanish - ProofWriter    (j) Spanish - FOLIO    (k) Spanish GSM8K    (l) Spanish - LogicalDeduction

(m) Hindi - ProofWriter    (n) Hindi - FOLIO    (o) Hindi GSM8K    (p) Hindi - LogicalDeduction

Figure 4: Performance of Llama-3-8B-Instruct across languages and datasets.

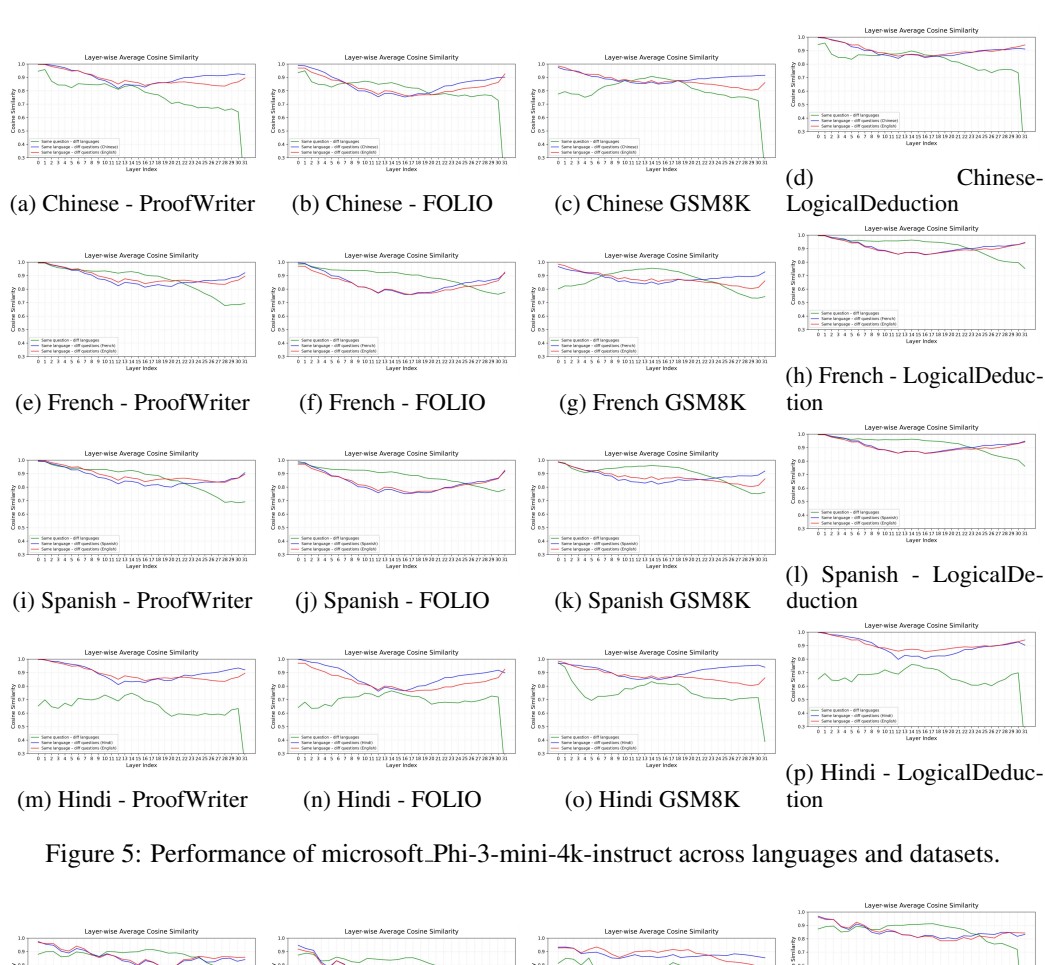

(a) Chinese - ProofWriter  (b) Chinese - FOLIO  (c) Chinese GSM8K  (d) Chinese-LogicalDeduction

(e) French - ProofWriter  (f) French - FOLIO  (g) French GSM8K  (h) French - LogicalDeduction

(i) Spanish - ProofWriter  (j) Spanish - FOLIO  (k) Spanish GSM8K  (l) Spanish - LogicalDeduction

(m) Hindi - ProofWriter  (n) Hindi - FOLIO  (o) Hindi GSM8K  (p) Hindi - LogicalDeduction

Figure 5: Performance of microsoft_Phi-3-mini-4k-instruct across languages and datasets.

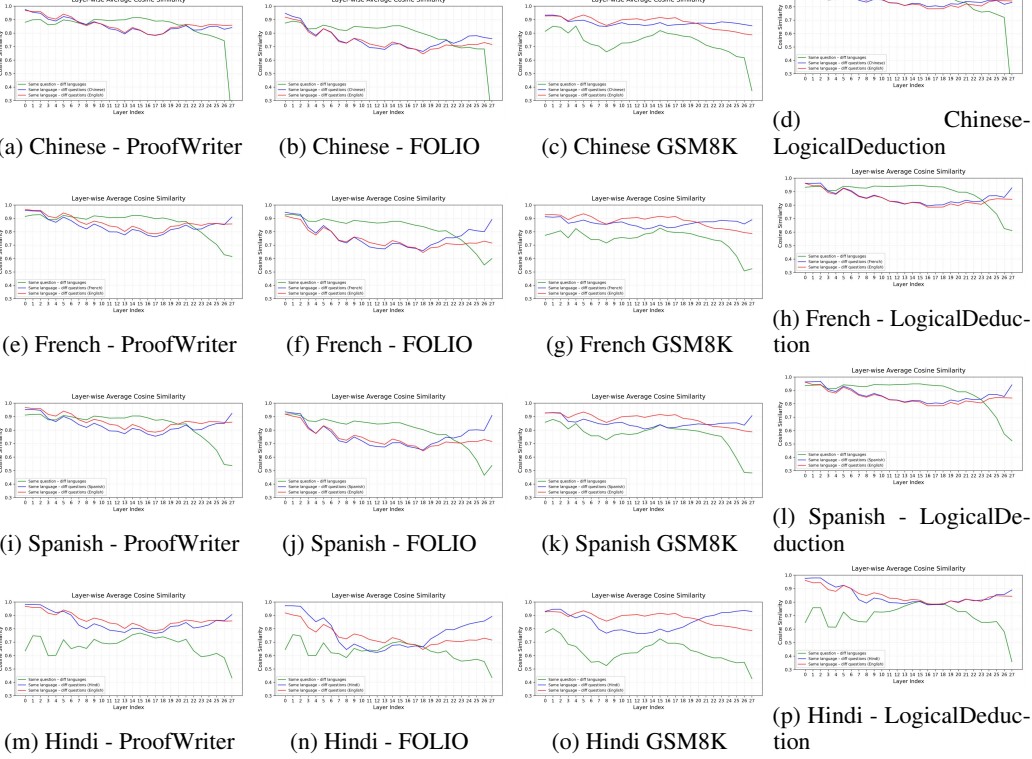

(a) Chinese - ProofWriter  (b) Chinese - FOLIO  (c) Chinese GSM8K  (d) Chinese-LogicalDeduction

(e) French - ProofWriter  (f) French - FOLIO  (g) French GSM8K  (h) French - LogicalDeduction

(i) Spanish - ProofWriter  (j) Spanish - FOLIO  (k) Spanish GSM8K  (l) Spanish - LogicalDeduction

(m) Hindi - ProofWriter  (n) Hindi - FOLIO  (o) Hindi GSM8K  (p) Hindi - LogicalDeduction

Figure 6: Performance of Qwen2-7B-Instruct across languages and datasets.

# D DETAILS OF COSINE SIMILARITY BETWEEN HIDDEN STATED IN ADJACENT LAYERS

In this section, we present the full details of the cosine similarity between hidden states in adjacent layers. We conduct experiments on three LLMs in subsection 3.1 across four datasets in subsection 3.2, which directly facilitate the identification of idle reasoning layers.

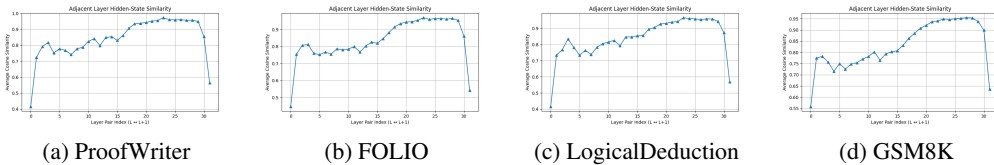

(a) ProofWriter  (b) FOLIO  (c) LogicalDeduction  (d) GSM8K

Figure 7: Cosine Similarity between Hidden Stated in Adjacent Layers of Llama-3-8B-Instruct

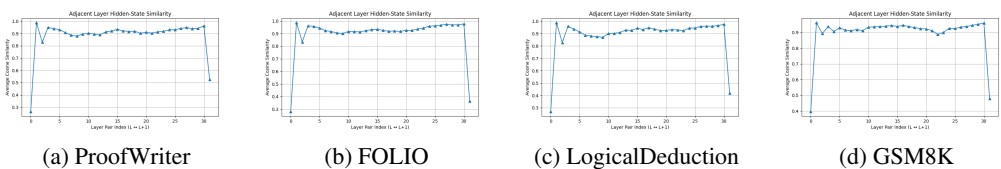

(a) ProofWriter  (b) FOLIO  (c) LogicalDeduction  (d) GSM8K

Figure 8: Cosine Similarity between Hidden Stated in Adjacent Layers of Phi-3-mini-4k-instruct

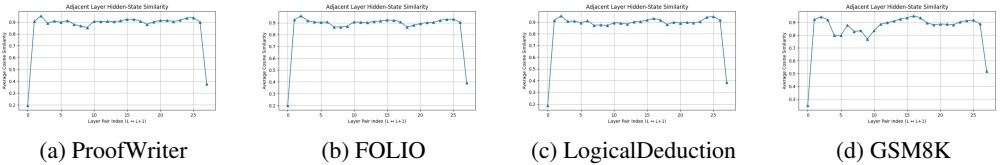

(a) ProofWriter  (b) FOLIO  (c) LogicalDeduction  (d) GSM8K

Figure 9: Cosine Similarity between Hidden Stated in Adjacent Layers of Qwen2-7B-Instruct

# E    RESULTS OF THE METHOD FOR FUNCTIONAL LAYER SEGMENTATION

In this Section, we provide a detailed explanation of the method used to approximate a reasonable location in Section 5, and introduce the values we ultimately selected for the functional layer segmentation.

Our method identifies the boundary between conceptualization and reasoning layers by finding the first significant inflection point in the cross-lingual task similarity curve ($g^k$). We locate this by calculating the discrete second derivative, $\Delta^2 g^k = g^{k+1} - 2g^k + g^{k-1}$, and identifying the first layer $k^*$ where the sign of $\Delta^2 g^k$ changes. To determine the boundary of idle reasoning layers, we simply select layers where the adjacent cosine similarity exceeds a predefined threshold $\beta$.

For instance, for the Llama-3-8B-Instruct model on the ProofWriter dataset (comparing Chinese and English versions), the layer-wise mean cosine similarities ($g^k$) are: [0.582, 0.640, 0.745, 0.772, 0.768, 0.780, 0.781, 0.797, 0.795, 0.812, 0.815, 0.824, 0.851, 0.841, 0.826, 0.826, 0.821, 0.811, 0.811, 0.799, 0.788, 0.793, 0.804, 0.813, 0.799, 0.748, 0.715, 0.677, 0.676, 0.629, 0.515, 0.357]

Calculating the discrete second derivative for this sequence, the first few values are:

- $\Delta^2 g^1 = g^2 - 2g^1 + g^0 \approx 0.745 - 2(0.640) + 0.582 = +\mathbf{0.047}$
- $\Delta^2 g^2 = g^3 - 2g^2 + g^1 \approx 0.772 - 2(0.745) + 0.640 = -\mathbf{0.078}$

The first sign change occurs at $k^* = 2$, which we identify as the boundary. We have performed this analysis across all our experimental settings and found this result to be remarkably consistent for this model. For the vast majority of tasks and languages, the first inflection point for Llama-3 is identified at **layer 2**. Based on this robust finding, we select layer 2 as the conceptualization-reasoning boundary for this model. In contrast, the boundary between active and idle reasoning layers is task-dependent; generally, more challenging tasks require a larger number of active reasoning layers. The specific segmentation results are summarized in the table below.

During the experiments, we observed that for certain datasets, the adjacent cosine similarity of Qwen2-7B-Instruct failed to exceed the threshold $\beta$. In such cases, we conclude that the task does not contain idle reasoning layers.

| Llama-3-8B-Instruct | | | | |
|---|---|---|---|---|
| Datasets | conceptualization layers | active reasoning layers | idle reasoning layers | textualization layers |
| ProofWriter | [0,2) | [2,20) | [20,30) | [30,32) |
| FOLIO | [0,2) | [2,20) | [20,30) | [30,32) |
| LogicalDeduction | [0,2) | [2,22) | [22,30) | [30,32) |
| GSM8K | [0,2) | [2,22) | [22,30) | [30,32) |

Table 4: Functional Layer Segmentation of Llama-3-8B-Instruct

| Phi-3-mini-4k-instruct | | | | |
|---|---|---|---|---|
| Datasets | conceptualization layers | active reasoning layers | idle reasoning layers | textualization layers |
| ProofWriter | [0,2) | [2,26) | [26,31) | [31,32) |
| FOLIO | [0,2) | [2,23) | [23,31) | [31,32) |
| LogicalDeduction | [0,2) | [2,24) | [24,31) | [31,32) |
| GSM8K | [0,2) | [2,28) | [28,31) | [31,32) |

Table 5: Functional Layer Segmentation of Phi-3-mini-4k-instruct

| Qwen2-7B-Instruct | | | | |
|---|---|---|---|---|
| Datasets | conceptualization layers | active reasoning layers | idle reasoning layers | textualization layers |
| ProofWriter | [0,3) | [3,26) | - | [26,28) |
| FOLIO | [0,3) | [3,26) | - | [26,28) |
| LogicalDeduction | [0,3) | [3,24) | [24,26) | [26,28) |
| GSM8K | [0,3) | [3,26) | - | [26,28) |

Table 6: Functional Layer Segmentation of Qwen2-7B-Instruct

## F    EXPERIMENTAL DETAILS

### F.0.1    TESTING METHOD

We test the finetuned models on their corresponding datasets. Since the reasoning datasets are multiple-choice questions, we can directly check whether the answers are correct.

For mathematical problems, we extract and compare the numerical answers. If an answer is not detected, we consider it incorrect by default.

In addition to evaluating the fine-tuning performance of LIFT and full Transformer layers, we also recorded the training time required for these fine-tuning tasks as another metric.

### F.1    EXPERIMENTAL ENVIRONMENT

Our experiments were conducted on a server equipped with **four NVIDIA A40 GPUs**, each with 46 GB of memory. All training and evaluation tasks were performed on a **single A40 GPU**. The server is powered by an **AMD EPYC 9654 96-Core Processor** with **755 GiB of RAM** and **16 TB of storage**.

We used **Ubuntu 20.04.6 LTS** as the operating system, with **Python 3.10.14** as the main programming environment. All deep learning experiments were implemented using **PyTorch 2.5.1+cu124**, leveraging CUDA for GPU acceleration.

### F.2    SOFTWARE AND LIBRARIES

- **CUDA Version**: 12.5
- **Driver Version**: 555.58.02
- **Deep Learning Framework**: PyTorch 2.5.1 with CUDA 12.4 support
- **Python Version**: 3.10.14
- **Operating System**: Ubuntu 20.04.6 LTS

### F.3    TRAINING CONFIGURATION

We fine-tuned these models using the following hyperparameters:

- **Learning Rate**: 1e-5
- **LoRA Rank**: 64
- **LoRA Alpha**: 128
- **LoRA Dropout**: 0.05
- **Batch Size per Device**: 4
- **Gradient Accumulation Steps**: 8
- **Max Sequence Length**: 512
- **Random Seeds**: 0, 1, 2, 3
- **Warmup Ratio**: 0.03

### F.4    EXECUTION STRATEGY

All training and evaluation were executed on a single A40 GPU to maintain consistency. Multi-GPU capabilities were available but not utilized for this experiment. Distributed strategies like FSDP or DeepSpeed were not applied in this setup.

## G    RESULTS OF ABLATION EXPERIMENTS

In this section, we present the results of our ablation experiments. Tables 7, 8, and 9 report the performance of fine-tuning various functional regions of different large language models (LLMs), including

the conceptualization layers, active reasoning layers, idle reasoning layers, and textualization layers. As discussed in Section 5, for the ProofWriter, FOLIO, and LogicalDeduction datasets, our LIFT algorithm fine-tunes the active reasoning layers. In contrast, for the GSM8K dataset, LIFT targets the conceptualization layers.

From these experiments, it is evident that our LIFT algorithm significantly outperforms competing approaches in most cases. The fine-tuning strategy, which focuses explicitly on functionally relevant layers, yields substantial improvements in task performance. These results highlight the critical role of selectively targeting specific functional layers within LLMs for effective task-specific optimization.

| Llama-3-8B-Instruct | | | | |
|---|---|---|---|---|
| Datasets | conceptualization layers | active reasoning layers | idle reasoning layers | textualization layers |
| ProofWriter | 37.25% (±1.04%) | 57.67% (±0.36%) | 48.08% (±0.75%) | 46.38% (±0.89%) |
| FOLIO | 53.43% (±1.65%) | 58.21% (±2.35%) | 56.74% (±1.29%) | 55.51% (±2.31%) |
| LogicalDeduction | 45.67% (±0.72%) | 53.58% (±2.47%) | 46.83% (±1.67%) | 45.25% (±0.88%) |
| GSM8K | 73.96% (±0.98%) | 67.87% (±0.64%) | 65.22% (±0.53%) | 56.48% (±0.58%) |

Table 7: Performance of Fine-Tuning over Various Layer Ranges of Llama-3-8B-Instruct

| Phi-3-mini-4k-instruct | | | | |
|---|---|---|---|---|
| Datasets | conceptualization layers | active reasoning layers | idle reasoning layers | textualization layers |
| ProofWriter | 62.42% (±0.67%) | 64.88% (±0.72%) | 53.42% (±1.50%) | 56.00% (±2.44%) |
| FOLIO | 61.40% (±1.46%) | 64.09% (±1.09%) | 51.96% (±1.79%) | 52.08% (±1.81%) |
| LogicalDeduction | 57.75% (±0.83%) | 61.75% (±2.49%) | 52.00% (±0.82%) | 45.08% (±1.23%) |
| GSM8K | 78.34% (±0.73%) | 76.82% (±0.25%) | 76.69% (±0.28%) | 79.57% (±0.41%) |

Table 8: Performance of Fine-Tuning over Various Layer Ranges of Phi-3-mini-4k-instruct

| Qwen2-7B-Instruct | | | | |
|---|---|---|---|---|
| Datasets | conceptualization layers | active reasoning layers | idle reasoning layers | textualization layers |
| ProofWriter | 45.50% (±0.93%) | 53.46% (±2.36%) | - | 47.46% (±0.85%) |
| FOLIO | 27.08% (±4.19%) | 44.73% (±1.98%) | - | 28.55% (±1.62%) |
| LogicalDeduction | 48.08% (±0.83%) | 57.08% (±3.29%) | 49.00% (±0.47%) | 52.58% (±2.15%) |
| GSM8K | 75.34% (±0.66%) | 75.78% (±0.24%) | - | 72.29% (±0.72%) |

Table 9: Performance of Fine-Tuning over Various Layer Ranges of Qwen2-7B-Instruct

# H  RESULTS OF VISUALIZATION OF HIDDEN STATES

In this section, we examine the internal representations of transformer-based models by visualizing the hidden states derived from different language versions of the same dataset. To facilitate this analysis, we employ three widely used dimensionality reduction techniques: Principal Component Analysis (PCA) (Hotelling, 1933), t-distributed Stochastic Neighbor Embedding (t-SNE) (Van der Maaten & Hinton, 2008), and Uniform Manifold Approximation and Projection (UMAP) (McInnes et al., 2018). For a controlled comparison, we utilize a single dataset with multiple languages, ensuring that the semantic content remains consistent across all versions. The model processes these multilingual inputs and generates corresponding hidden state vectors.

The visualizations produced by PCA, t-SNE, and UMAP (shown in the following figure) provide a robust basis for analyzing the hidden state representations across different languages. Our results reveal a consistent pattern in the evolution of hidden states across the layers of the language model. Specifically, in the input-adjacent and output-adjacent layers, internal representations tend to form clusters based on language. In contrast, in the intermediate layers, hidden states from different languages become increasingly intermixed, exhibiting a clear phenomenon of language blending. This suggests that while early and late layers are more language-specific, intermediate layers abstract away from surface linguistic forms and encode more universal semantic representations.

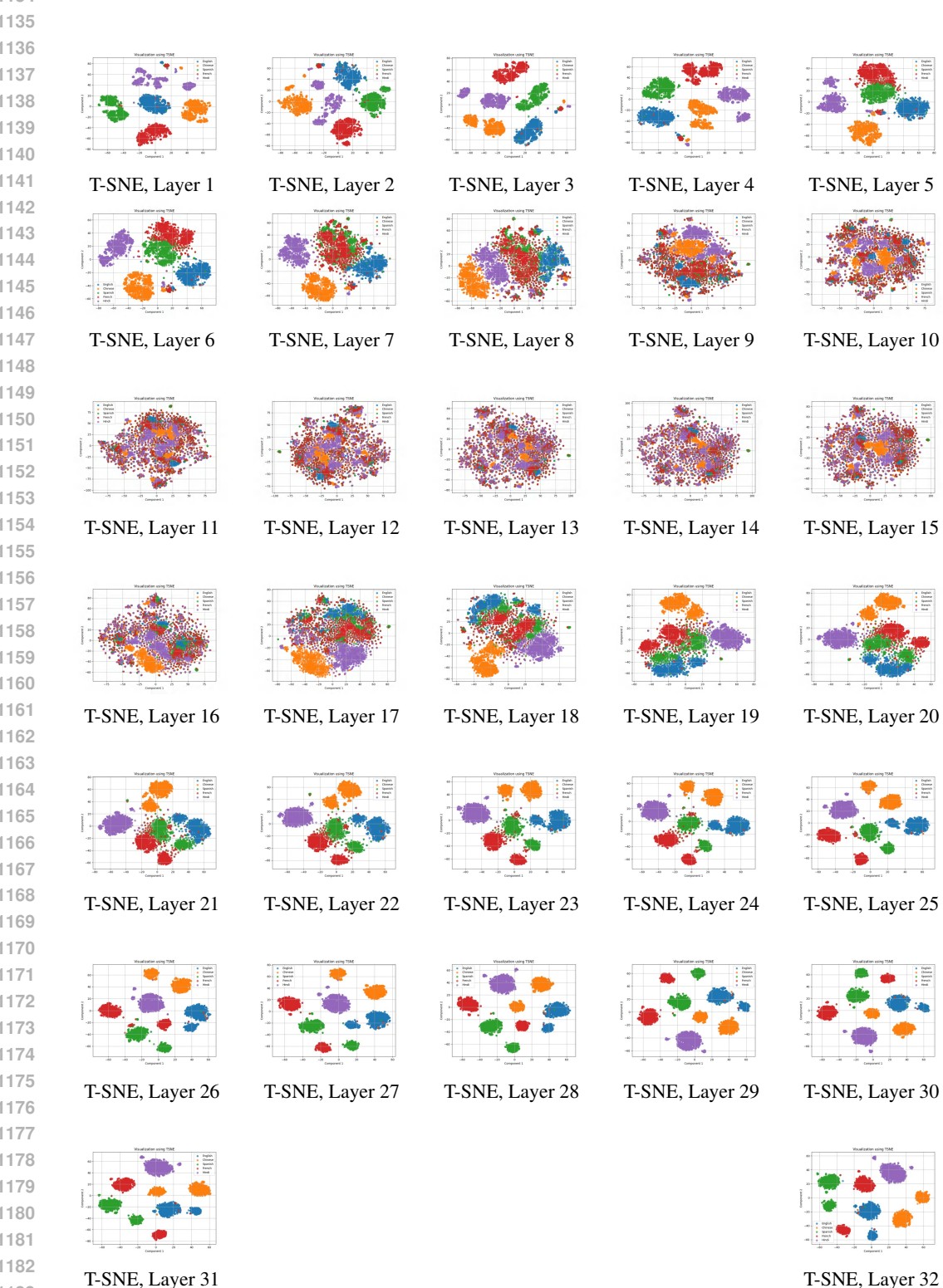

Figure 10: T-SNE visualizations for layers 1-32 of Llama-3-8B-Instruct on the GSM8K dataset.

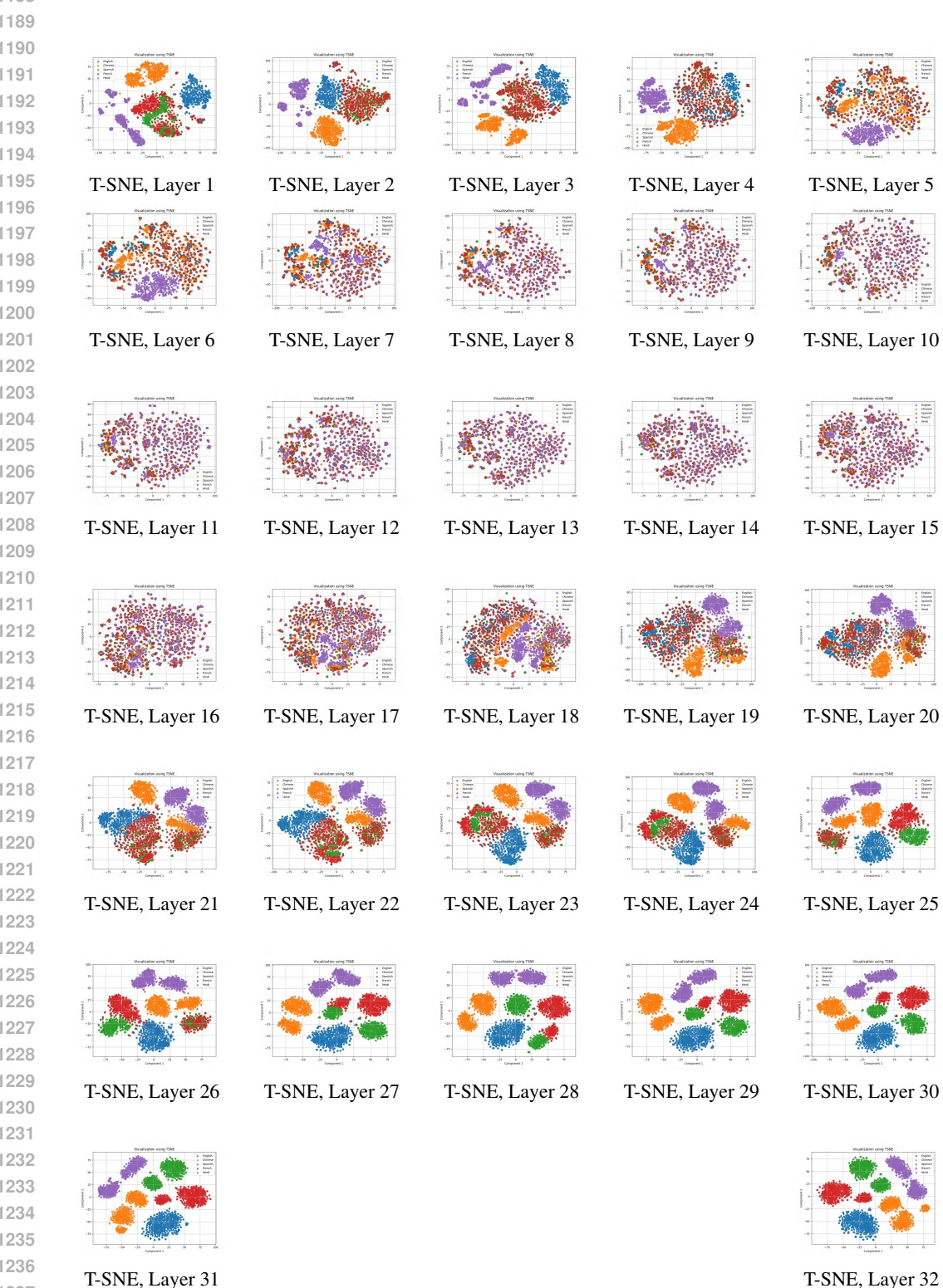

Figure 11: T-SNE visualizations for layers 1-32 of Llama-3-8B-Instruct on the FOLIO dataset.

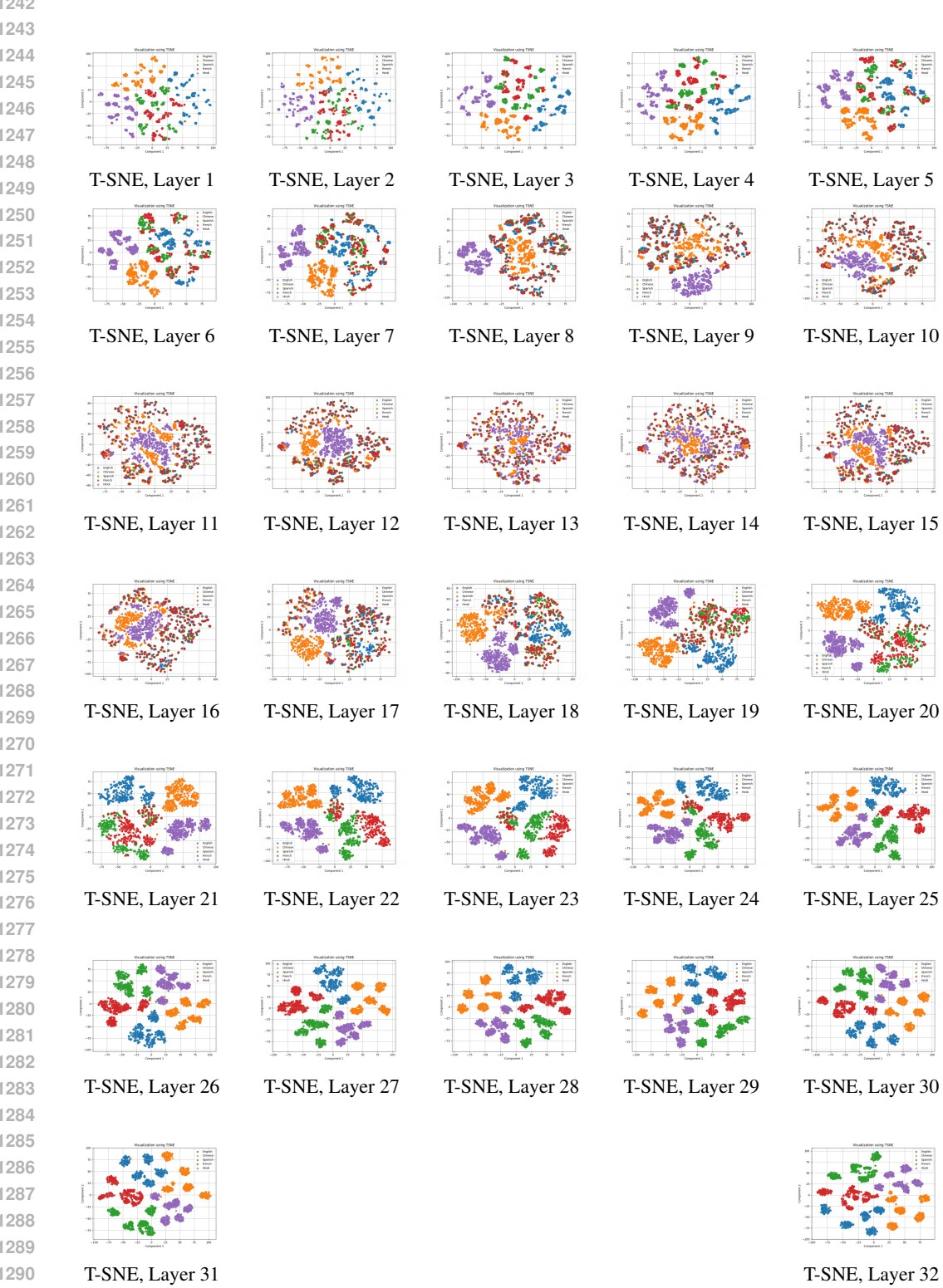

Figure 12: T-SNE visualizations for layers 1-32 of Llama-3-8B-Instruct on the LogicalDeduction dataset.

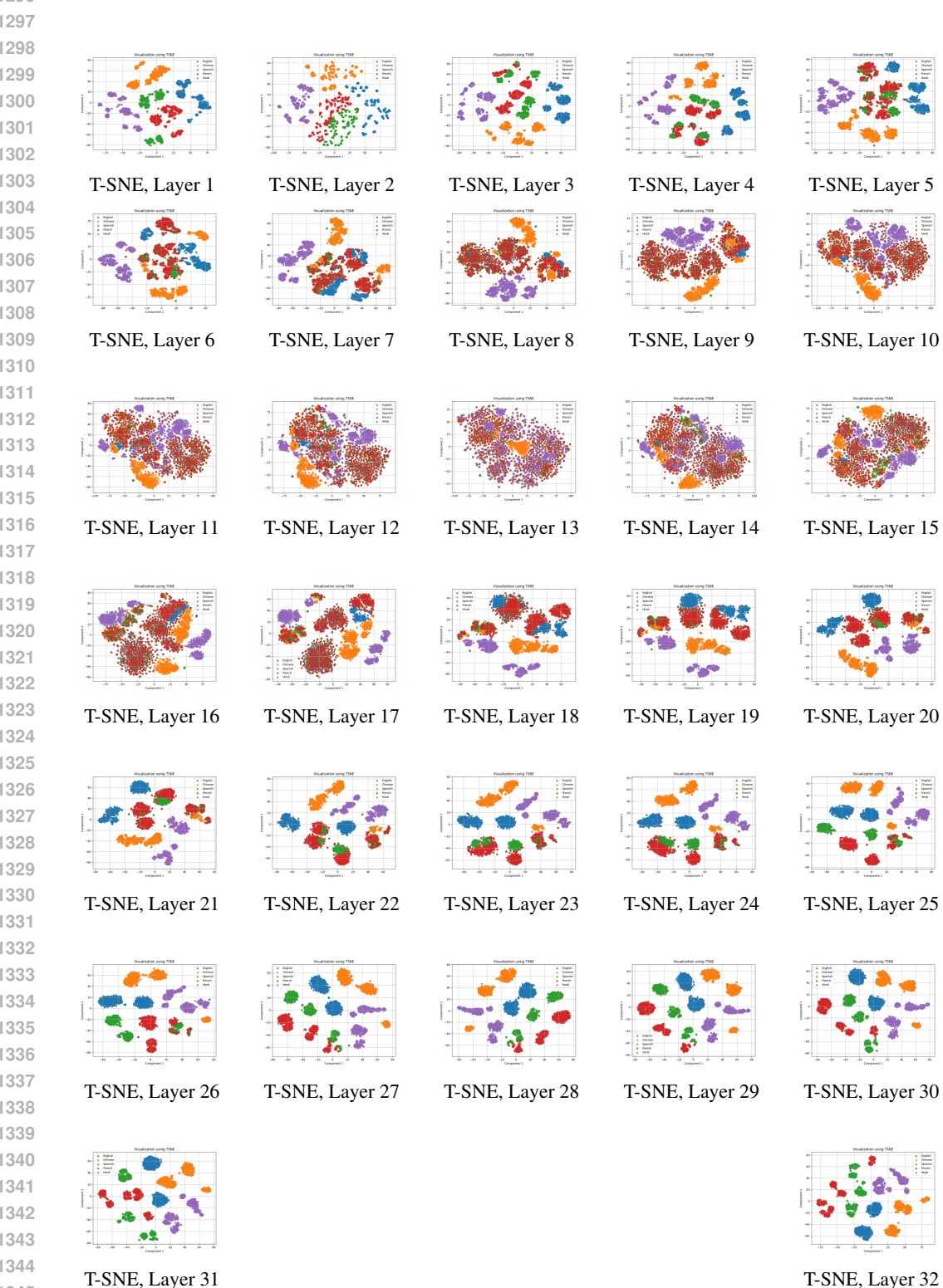

Figure 13: T-SNE visualizations for layers 1-32 of Llama-3-8B-Instruct on the ProofWriter dataset.

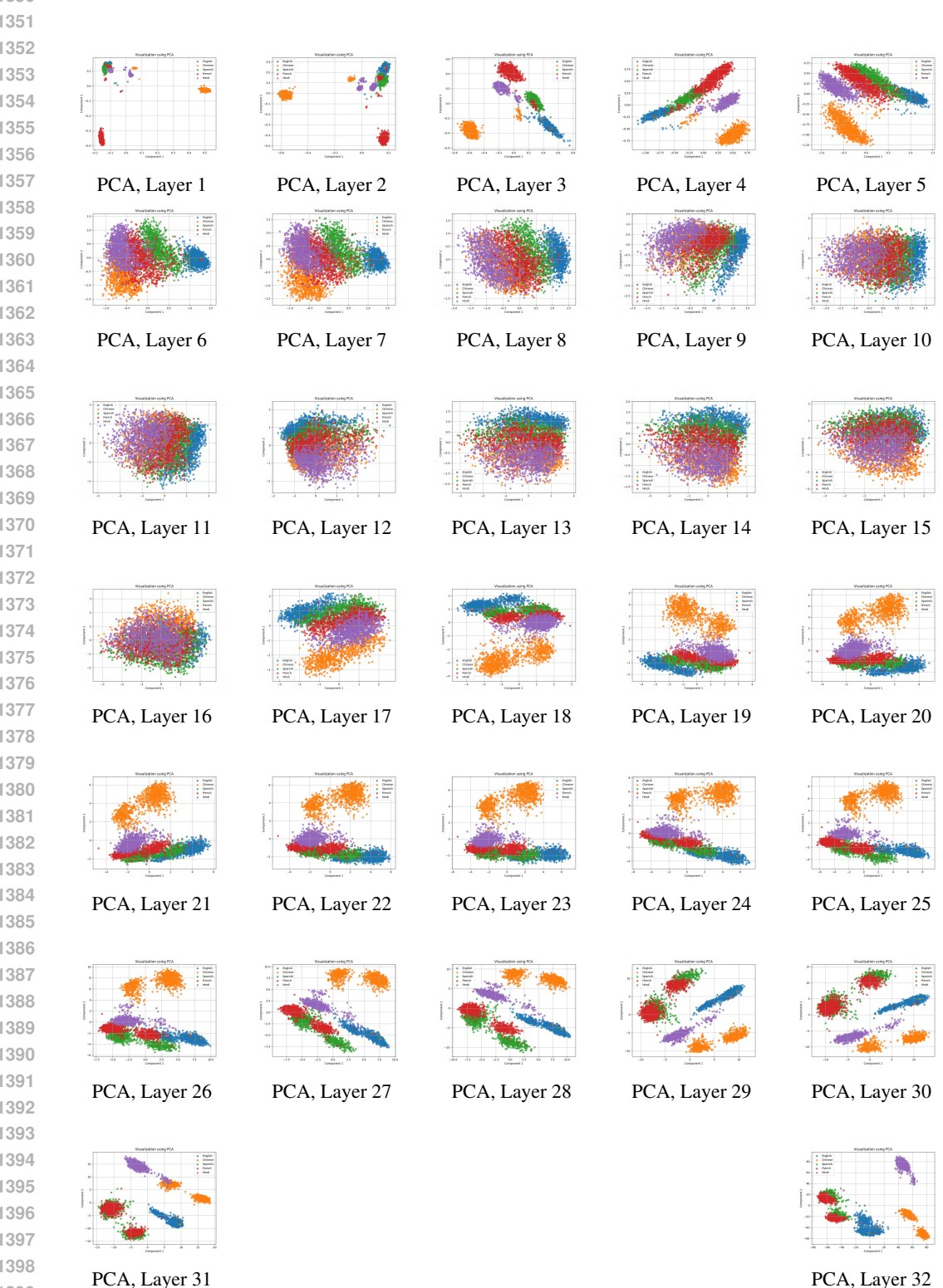

Figure 14: PCA visualizations for layers 1-32 of Llama-3-8B-Instruct on the GSM8K dataset.

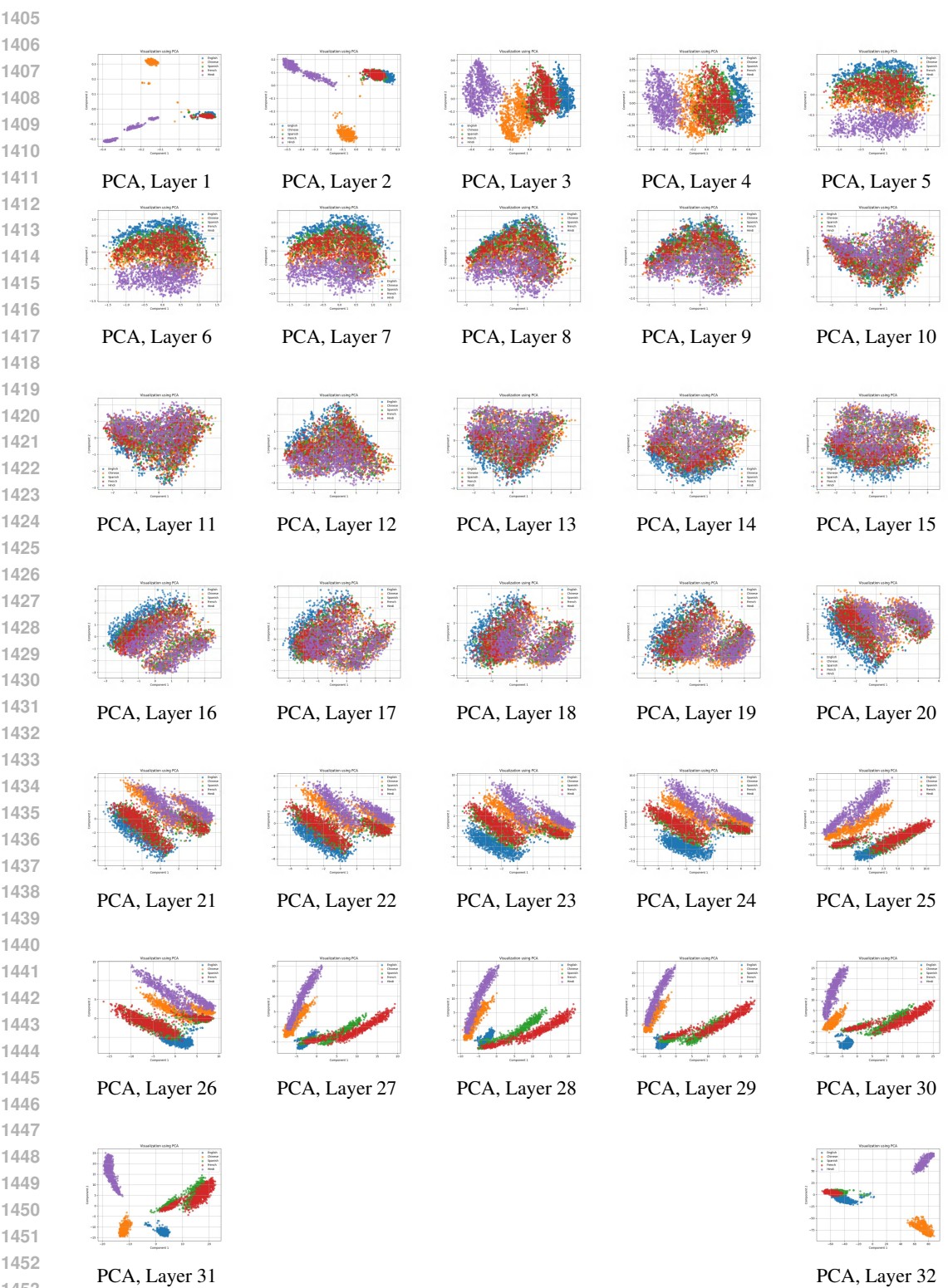

Figure 15: PCA visualizations for layers 1-32 of Llama-3-8B-Instruct on the FOLIO dataset.

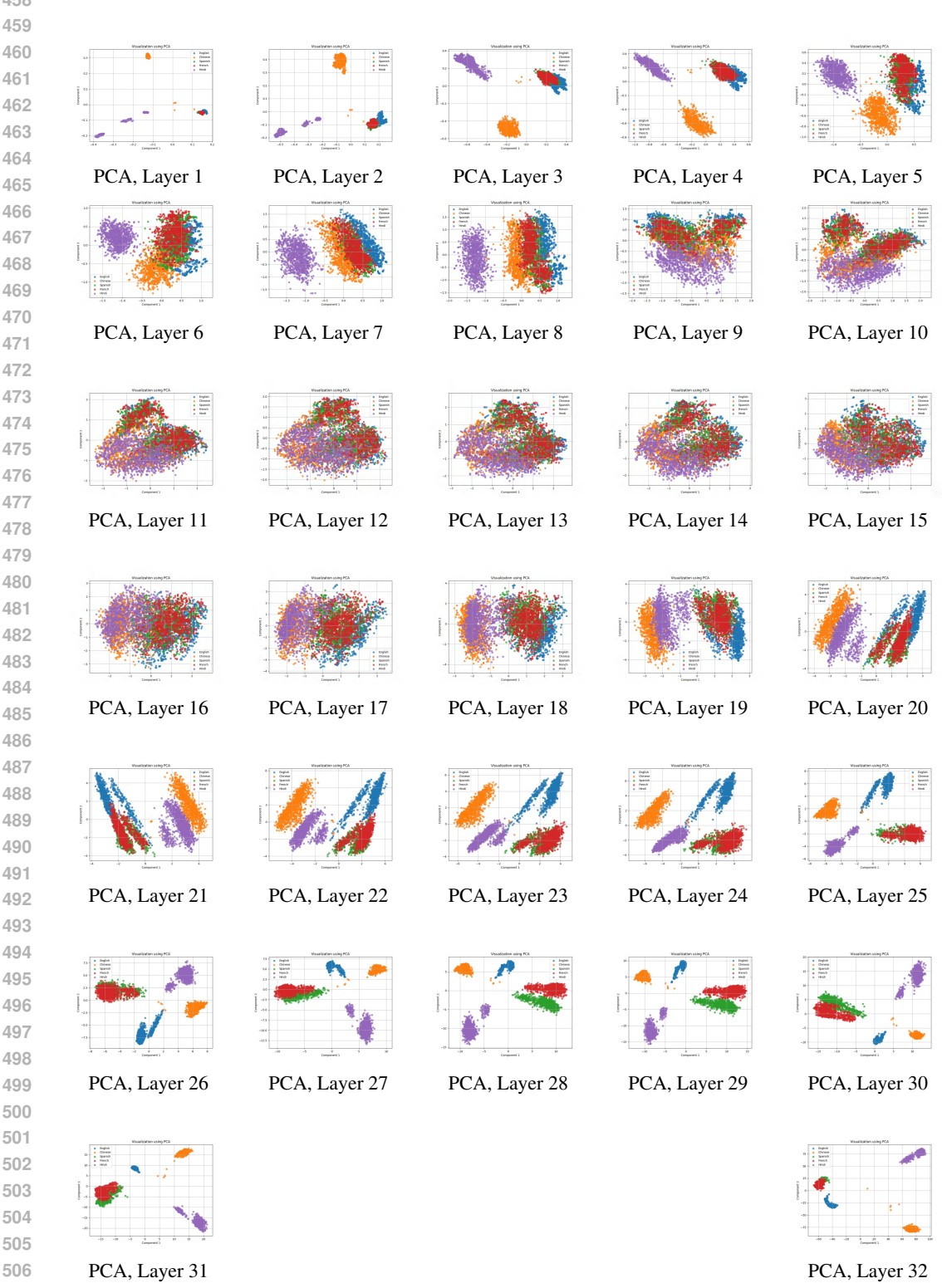

Figure 16: PCA visualizations for layers 1-32 of Llama-3-8B-Instruct on the LogicalDeduction dataset.

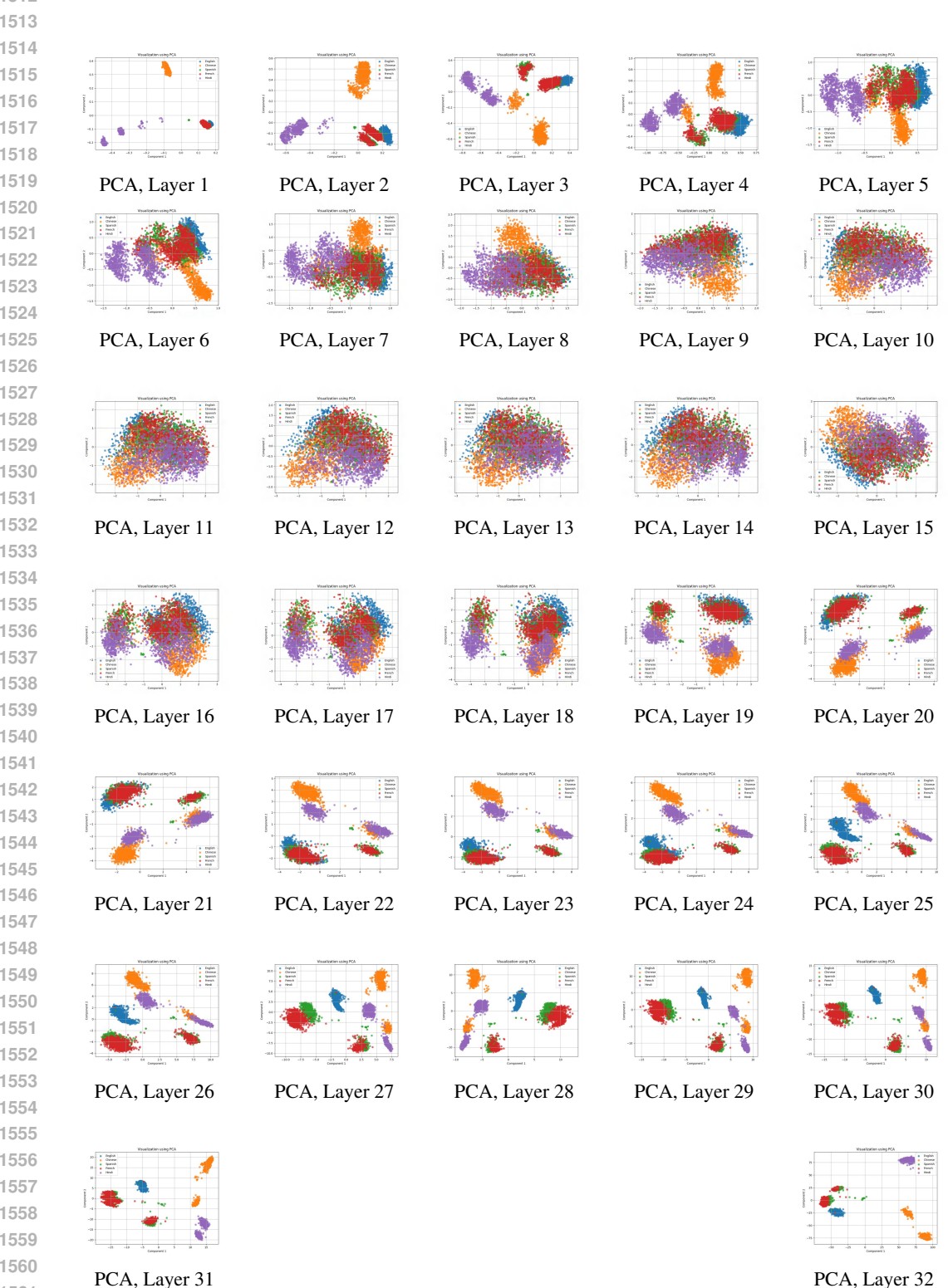

Figure 17: PCA visualizations for layers 1-32 of Llama-3-8B-Instruct on the ProofWriter dataset.

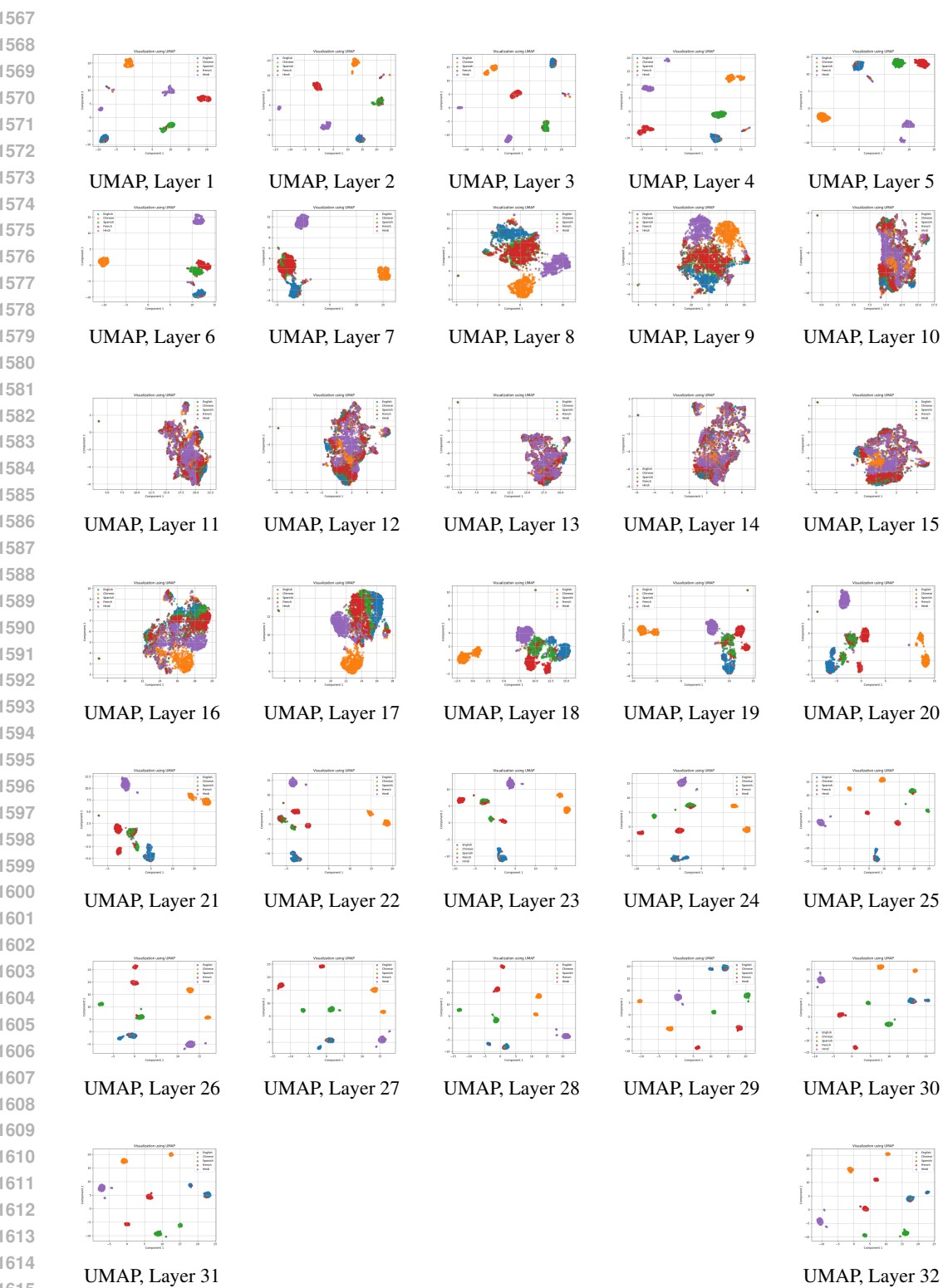

Figure 18: UMAP visualizations for layers 1-32 of Llama-3-8B-Instruct on the GSM8K dataset.

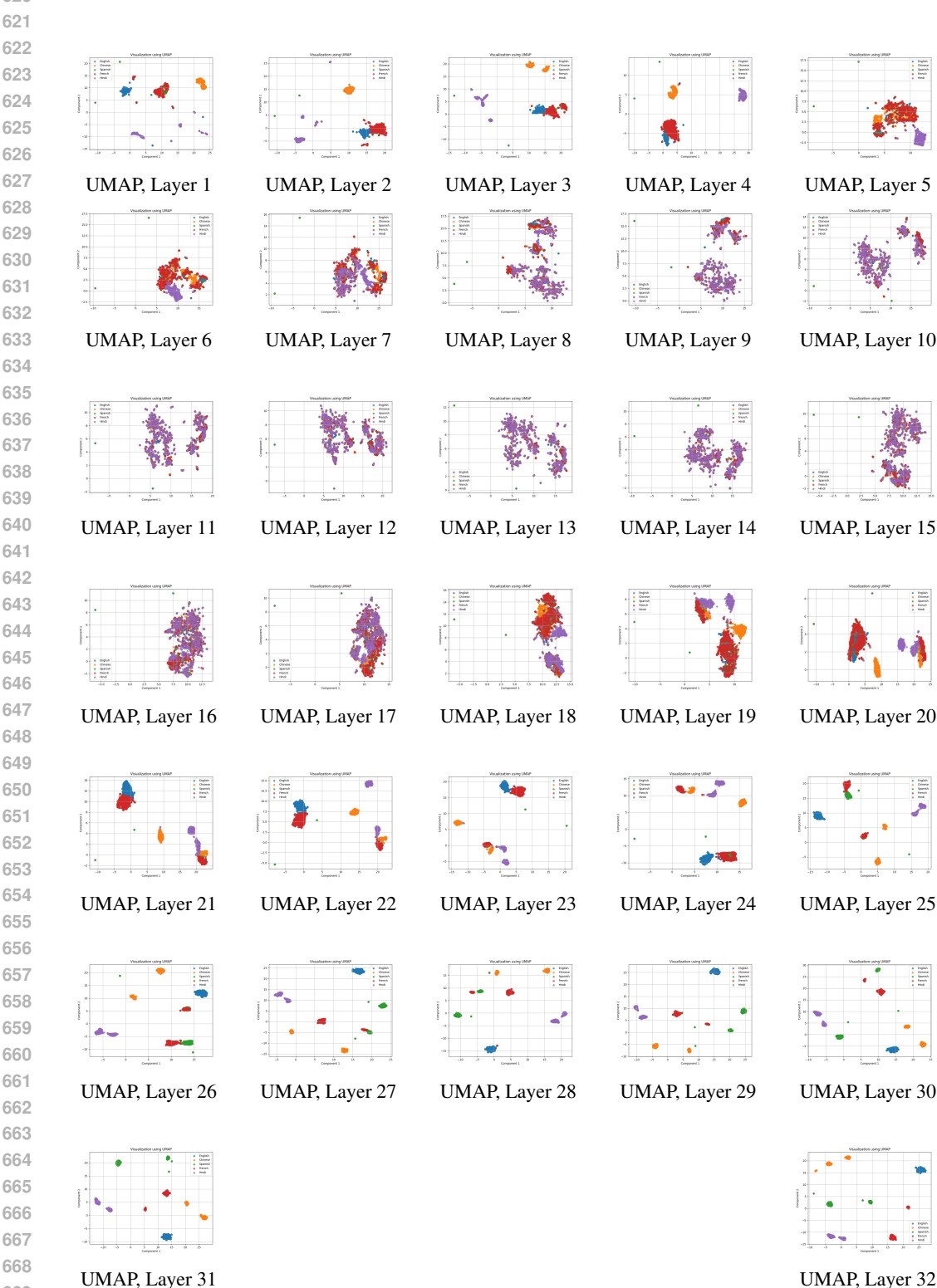

Figure 19: UMAP visualizations for layers 1-32 of Llama-3-8B-Instruct on the FOLIO dataset.

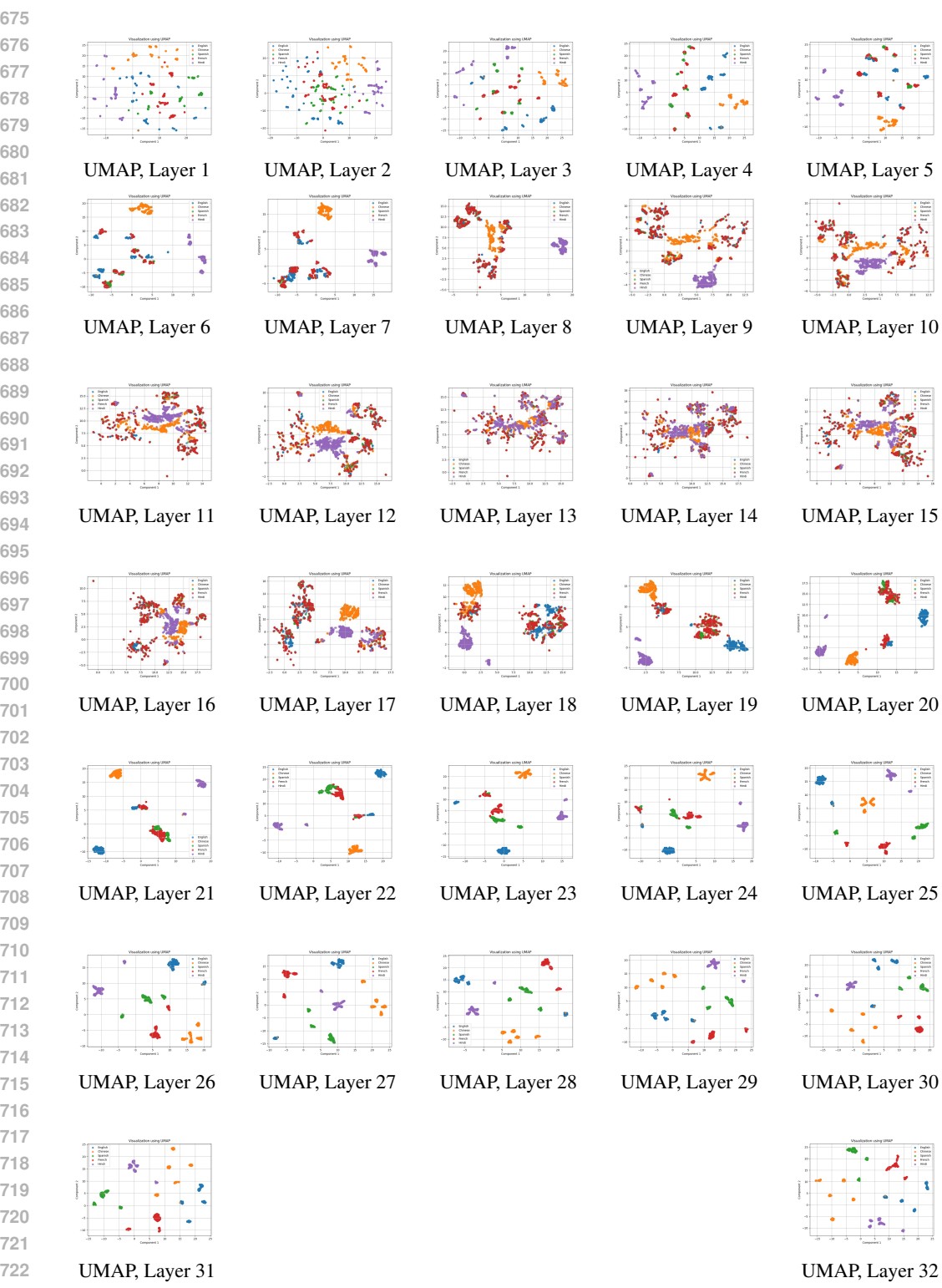

Figure 20: UMAP visualizations for layers 1-32 of Llama-3-8B-Instruct on the LogicalDeduction dataset.

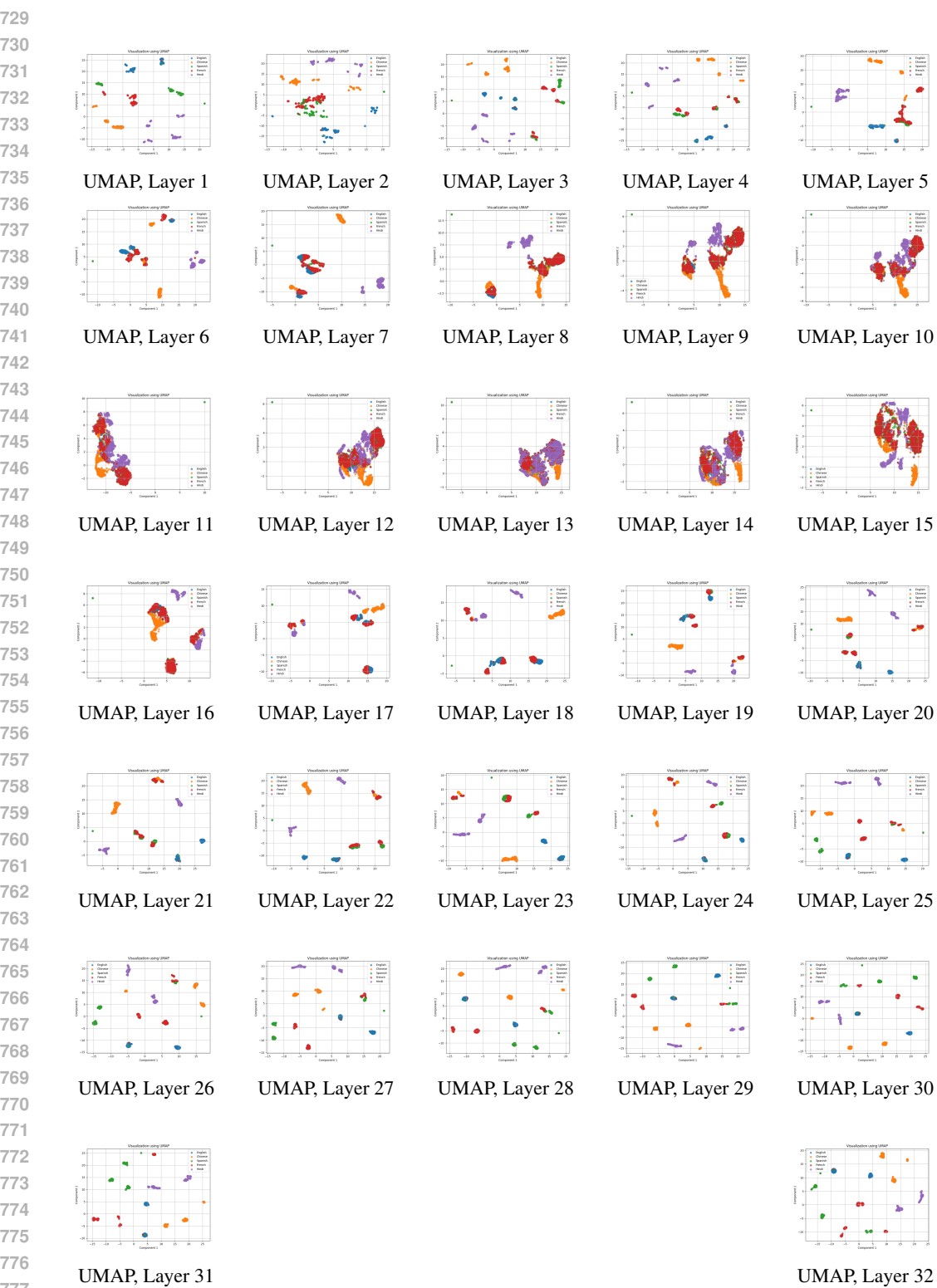

Figure 21: UMAP visualizations for layers 1-32 of Llama-3-8B-Instruct on the ProofWriter dataset.

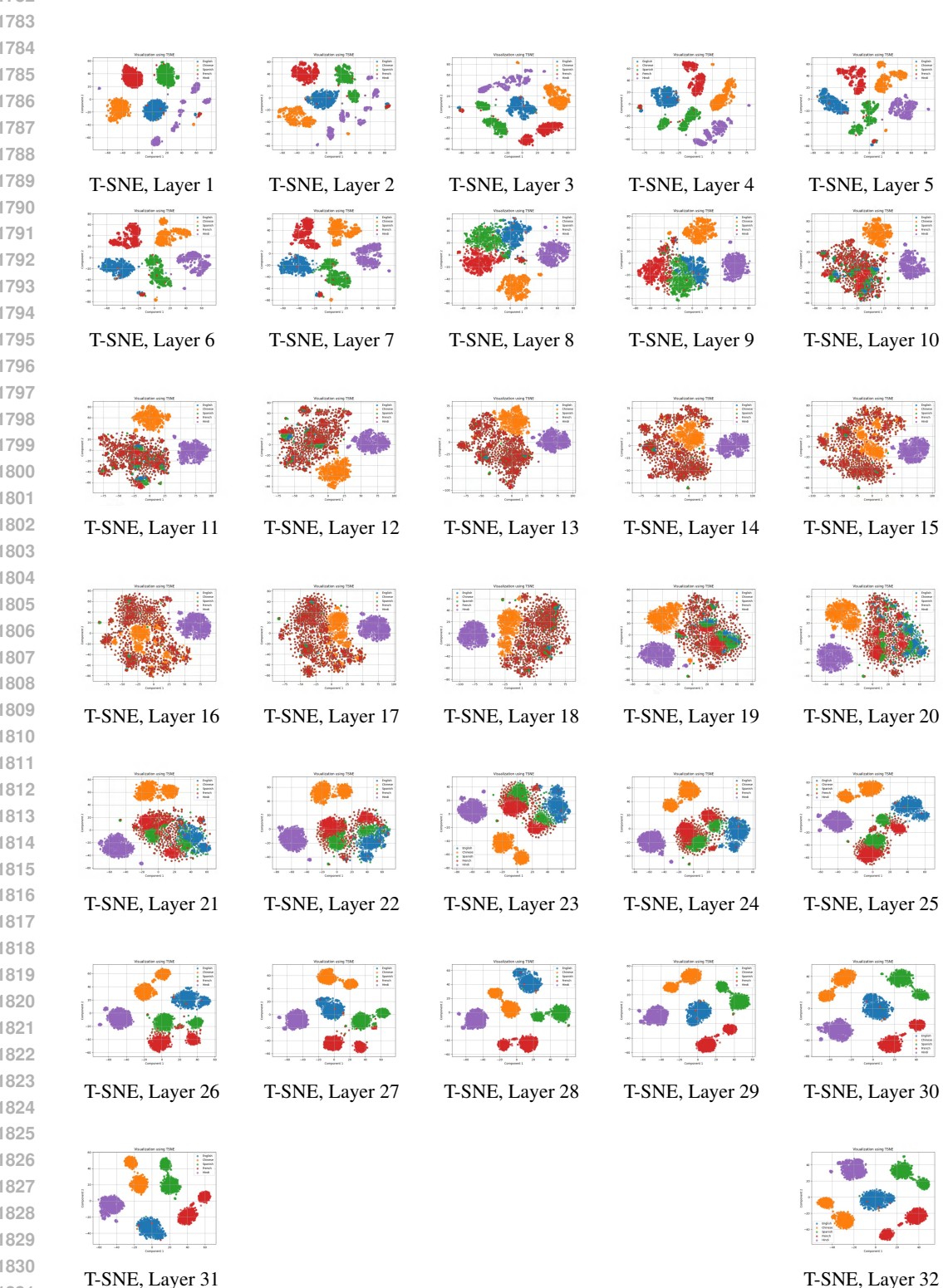

Figure 22: T-SNE visualizations for layers 1-32 of Phi-3-mini-4k-instruct on the GSM8K dataset.

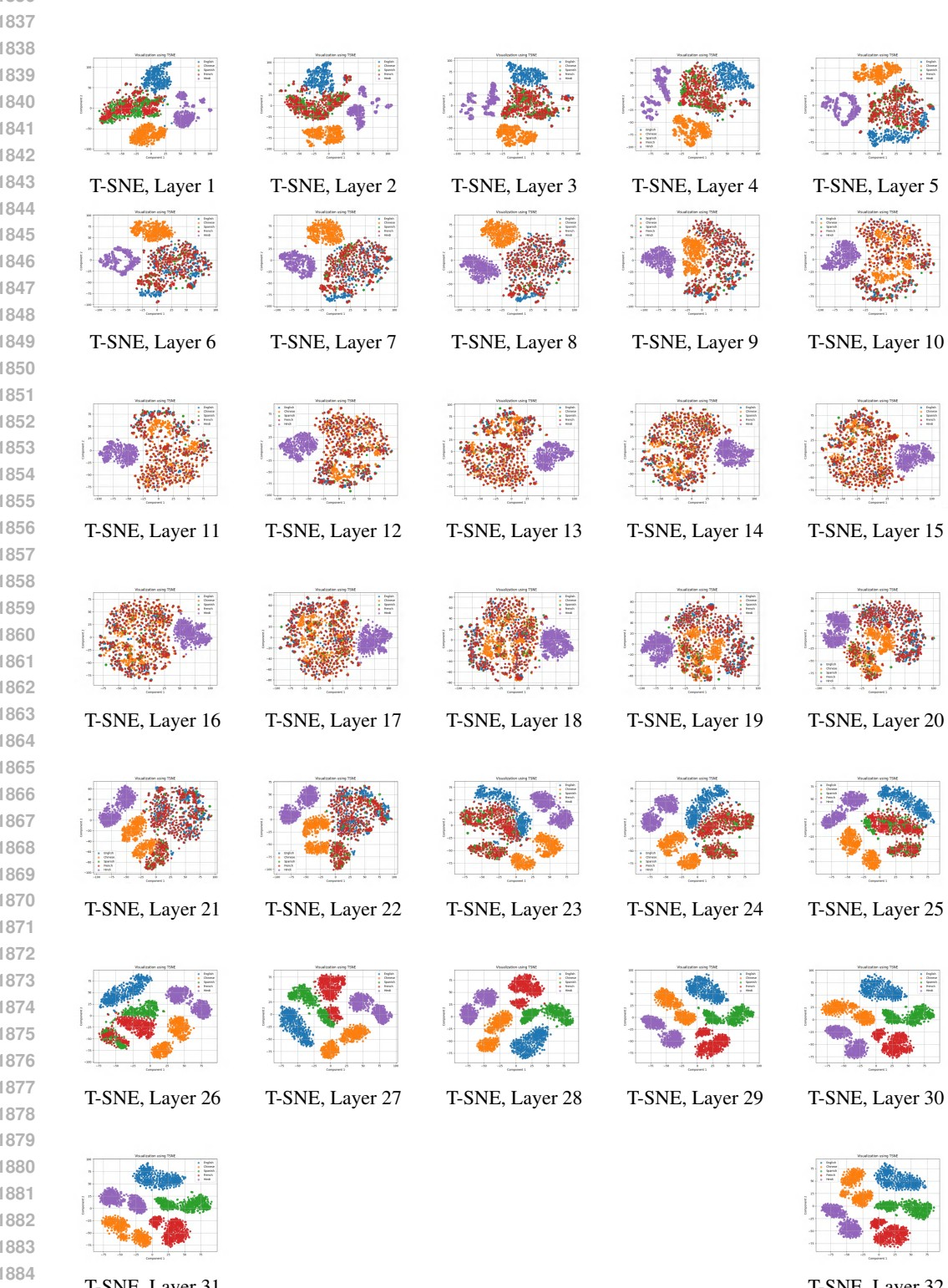

Figure 23: T-SNE visualizations for layers 1-32 of Phi-3-mini-4k-instruct on the FOLIO dataset.

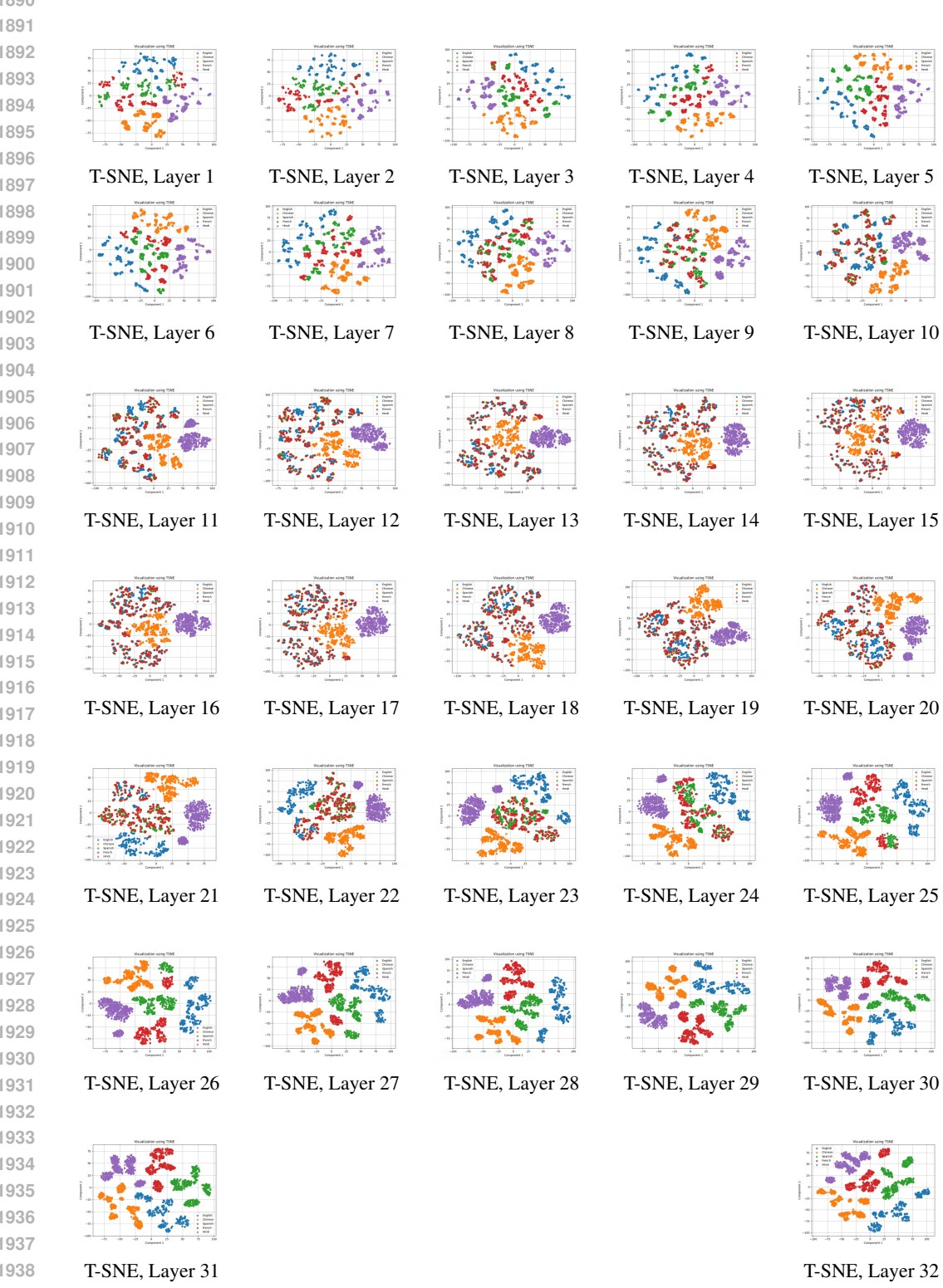

Figure 24: T-SNE visualizations for layers 1-32 of Phi-3-mini-4k-instruct on the LogicalDeduction dataset.

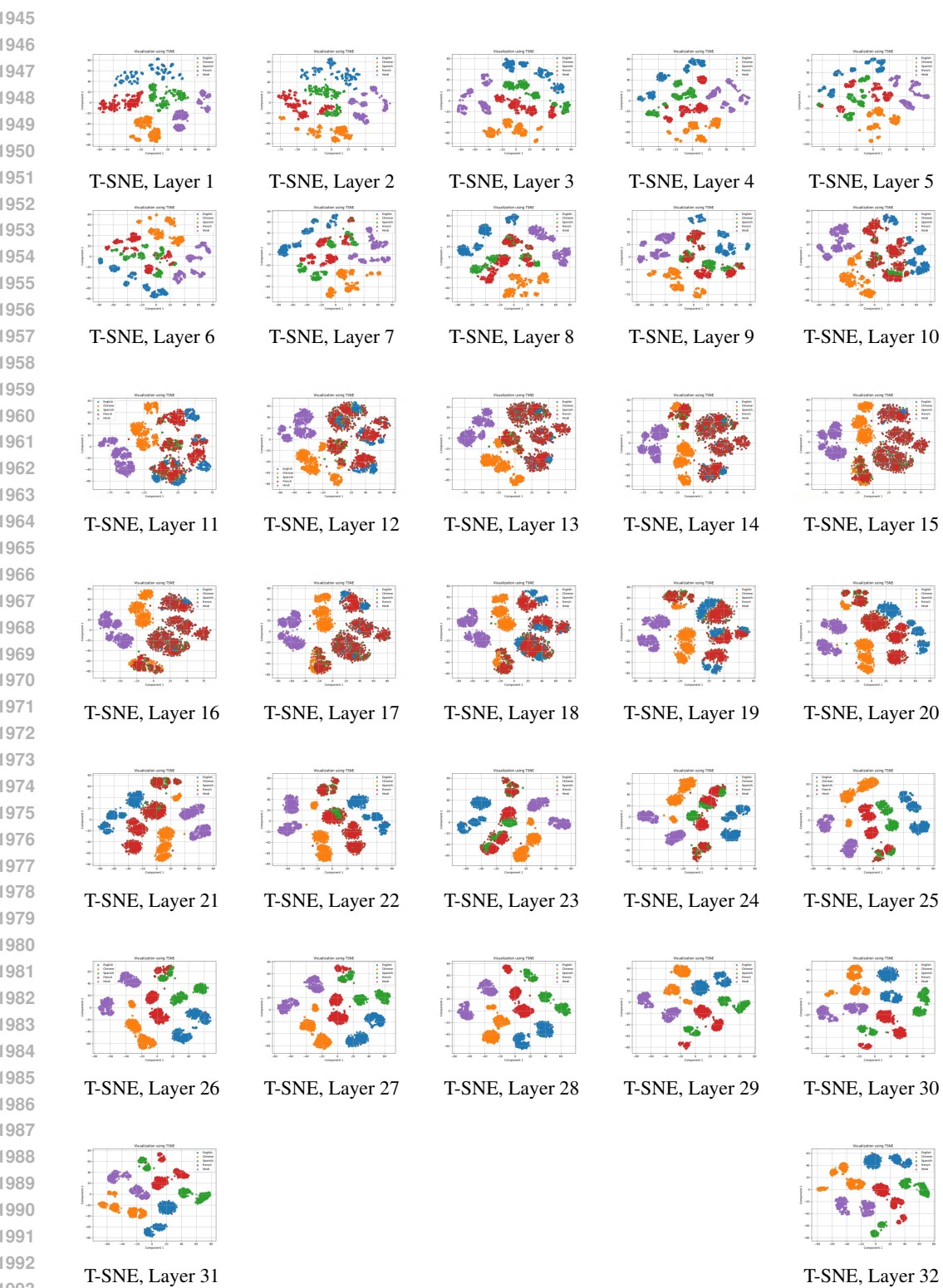

Figure 25: T-SNE visualizations for layers 1-32 of Phi-3-mini-4k-instruct on the ProofWriter dataset.

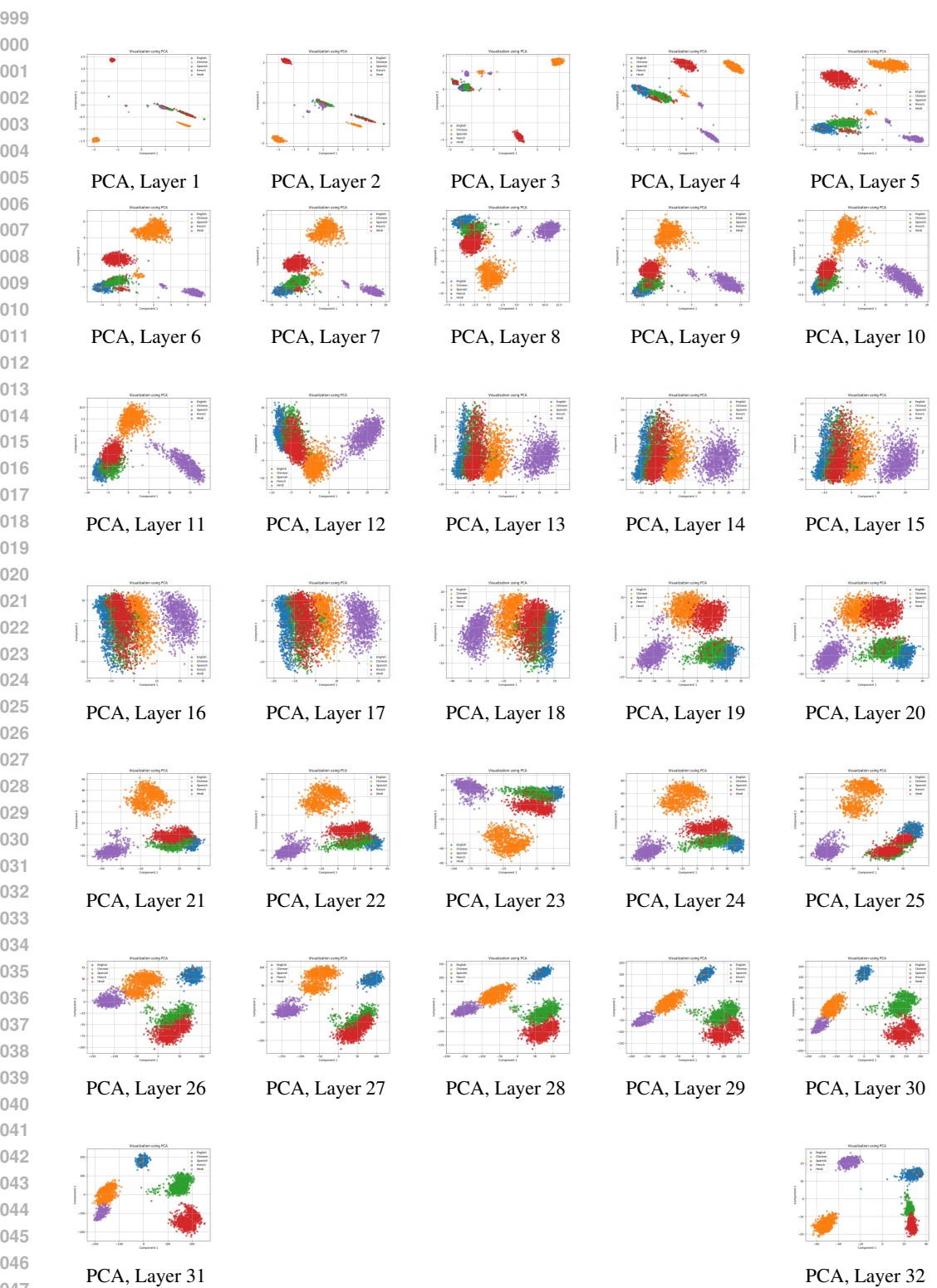

Figure 26: PCA visualizations for layers 1-32 of Phi-3-mini-4k-instruct on the GSM8K dataset.

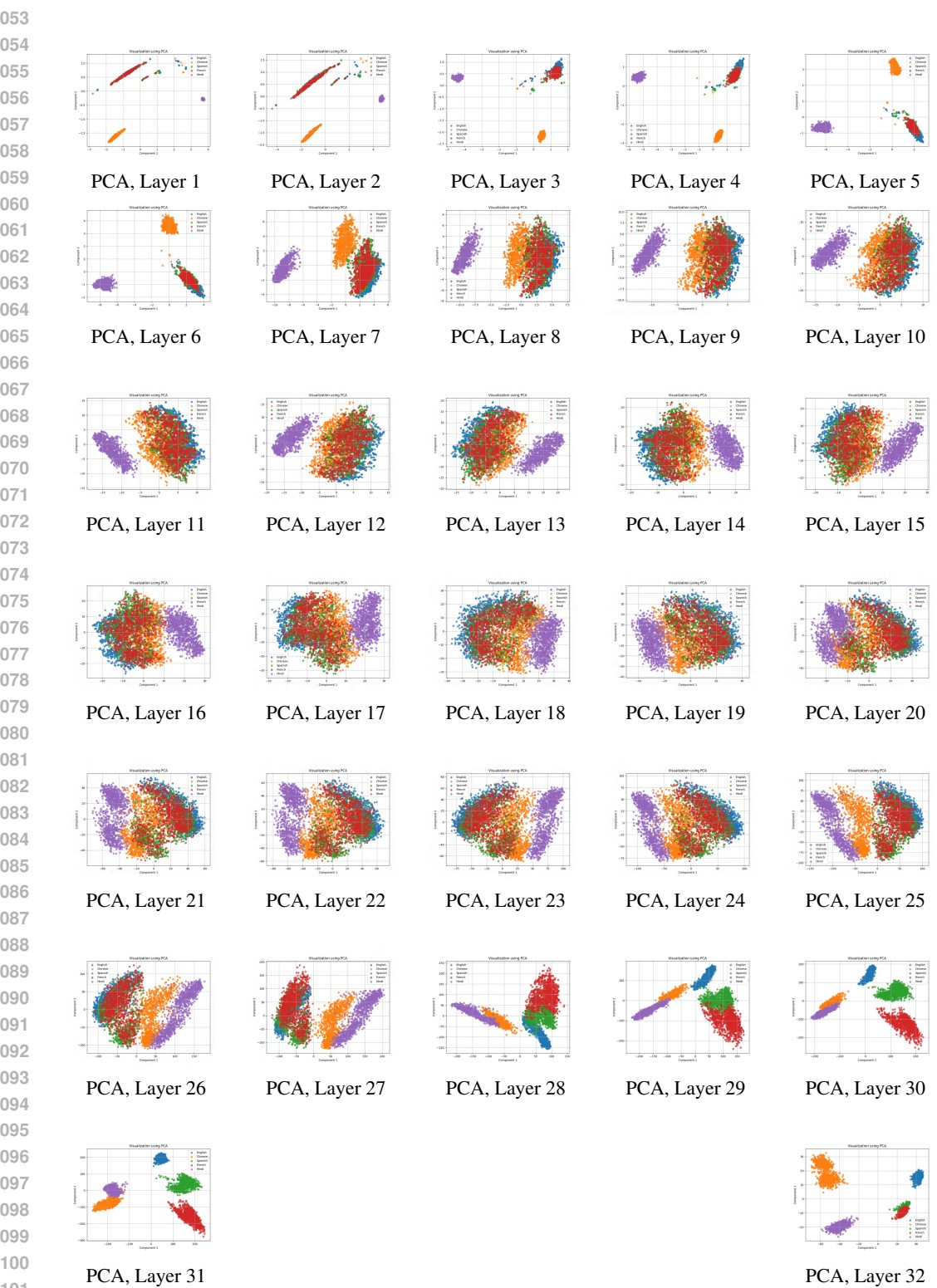

Figure 27: PCA visualizations for layers 1-32 of Phi-3-mini-4k-instruct on the FOLIO dataset.

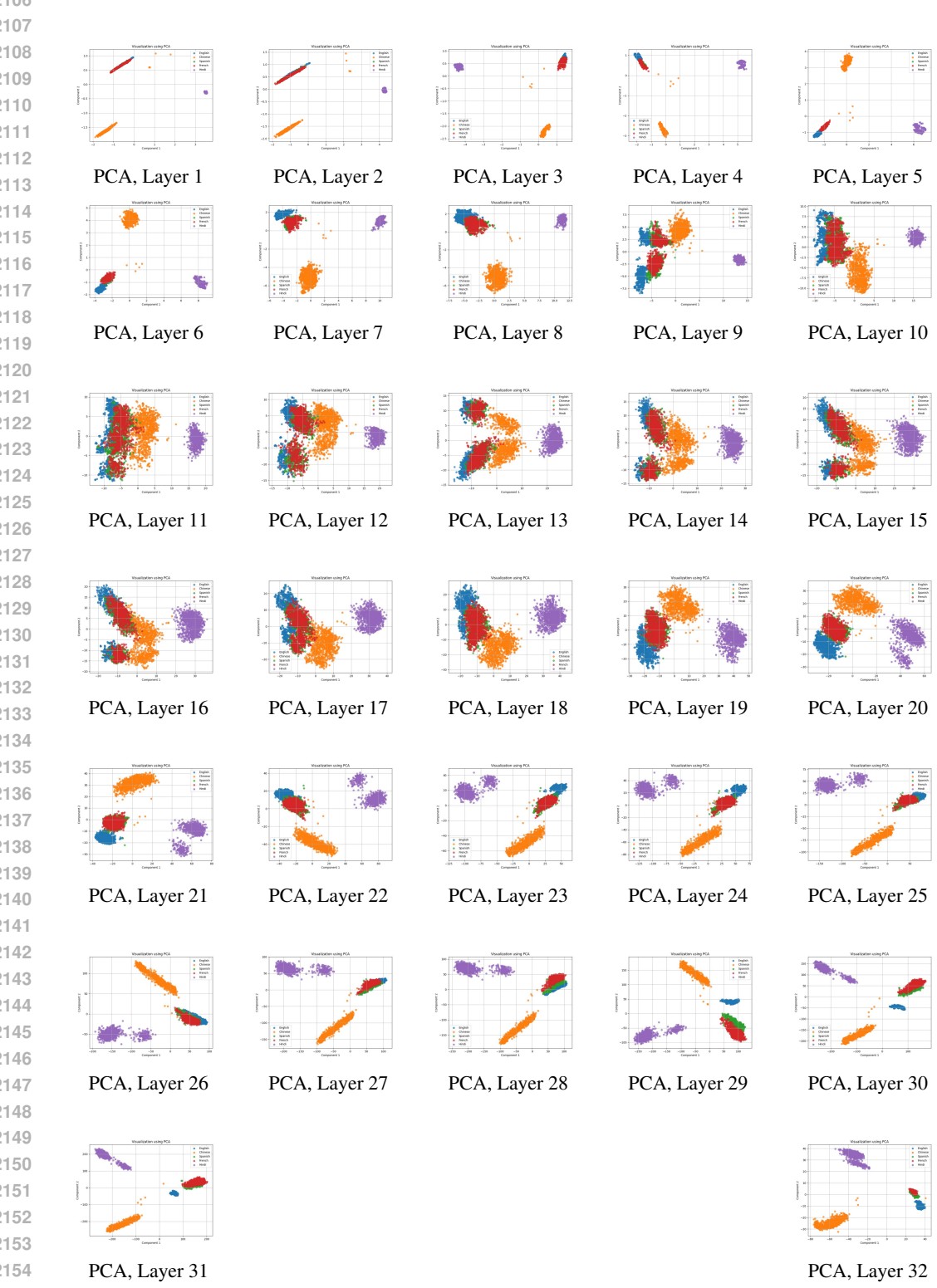

Figure 28: PCA visualizations for layers 1-32 of Phi-3-mini-4k-instruct on the LogicalDeduction dataset.

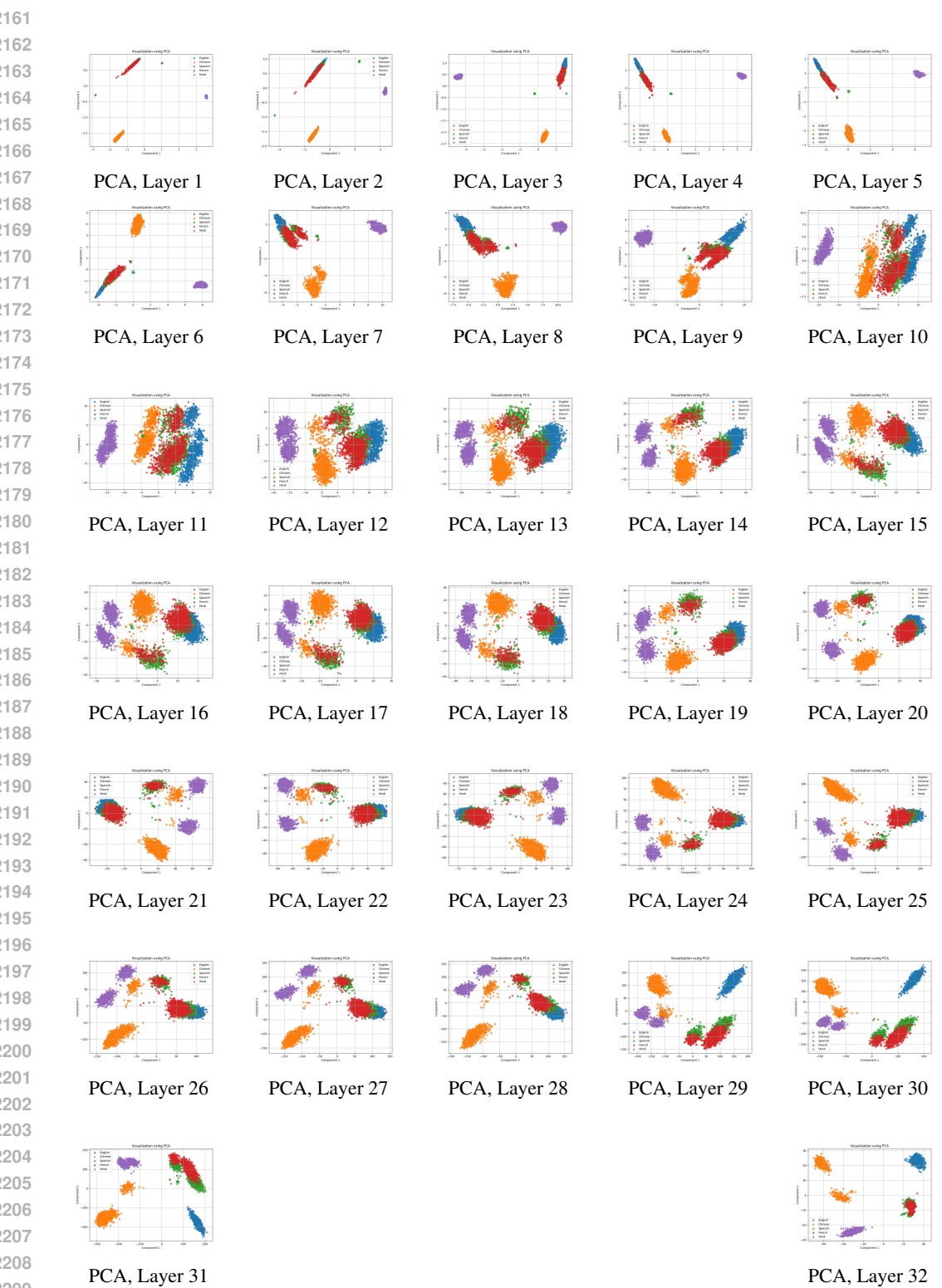

Figure 29: PCA visualizations for layers 1-32 of Phi-3-mini-4k-instruct on the ProofWriter dataset.

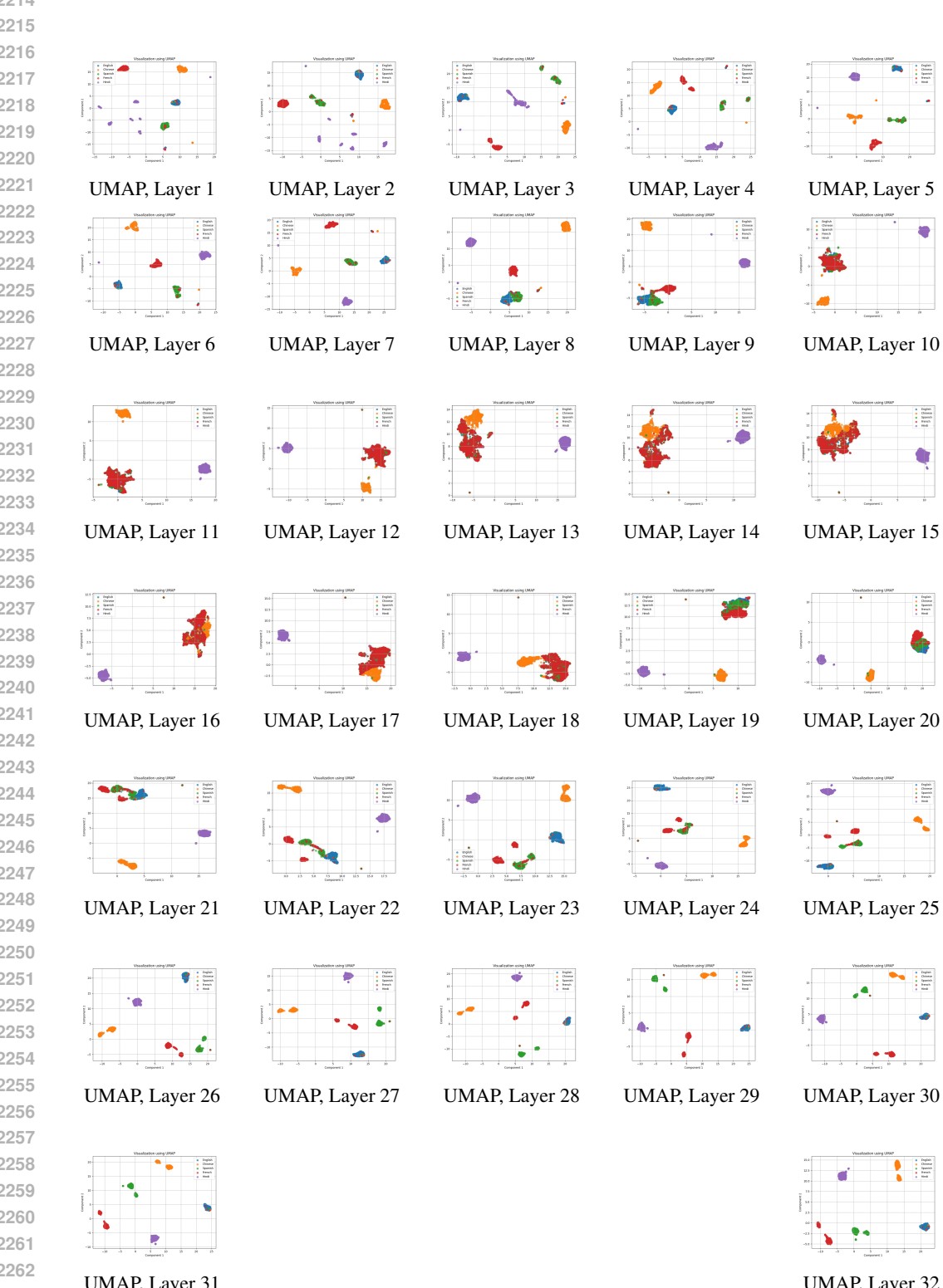

Figure 30: UMAP visualizations for layers 1-32 of Phi-3-mini-4k-instruct on the GSM8K dataset.

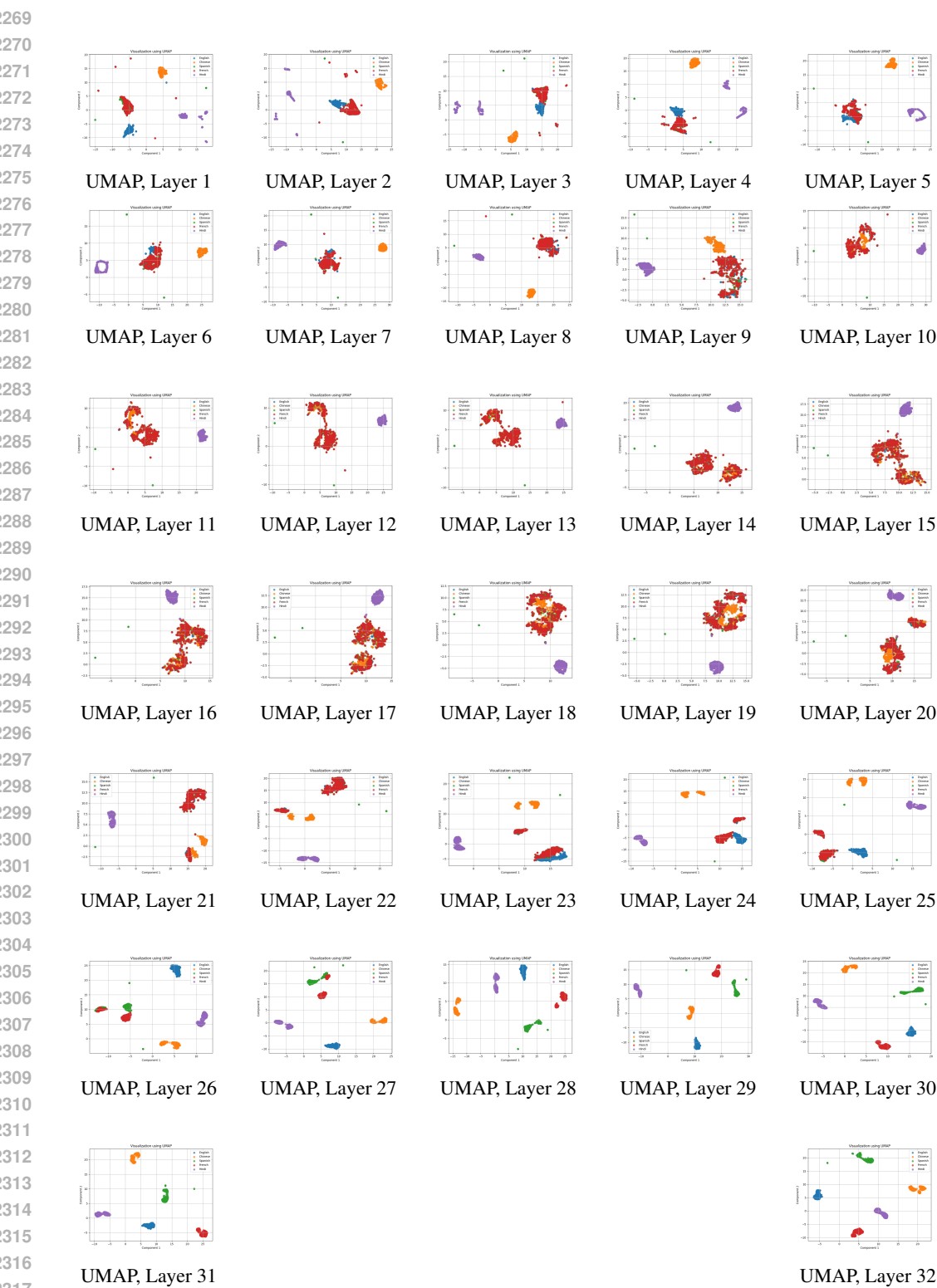

Figure 31: UMAP visualizations for layers 1-32 of Phi-3-mini-4k-instruct on the FOLIO dataset.

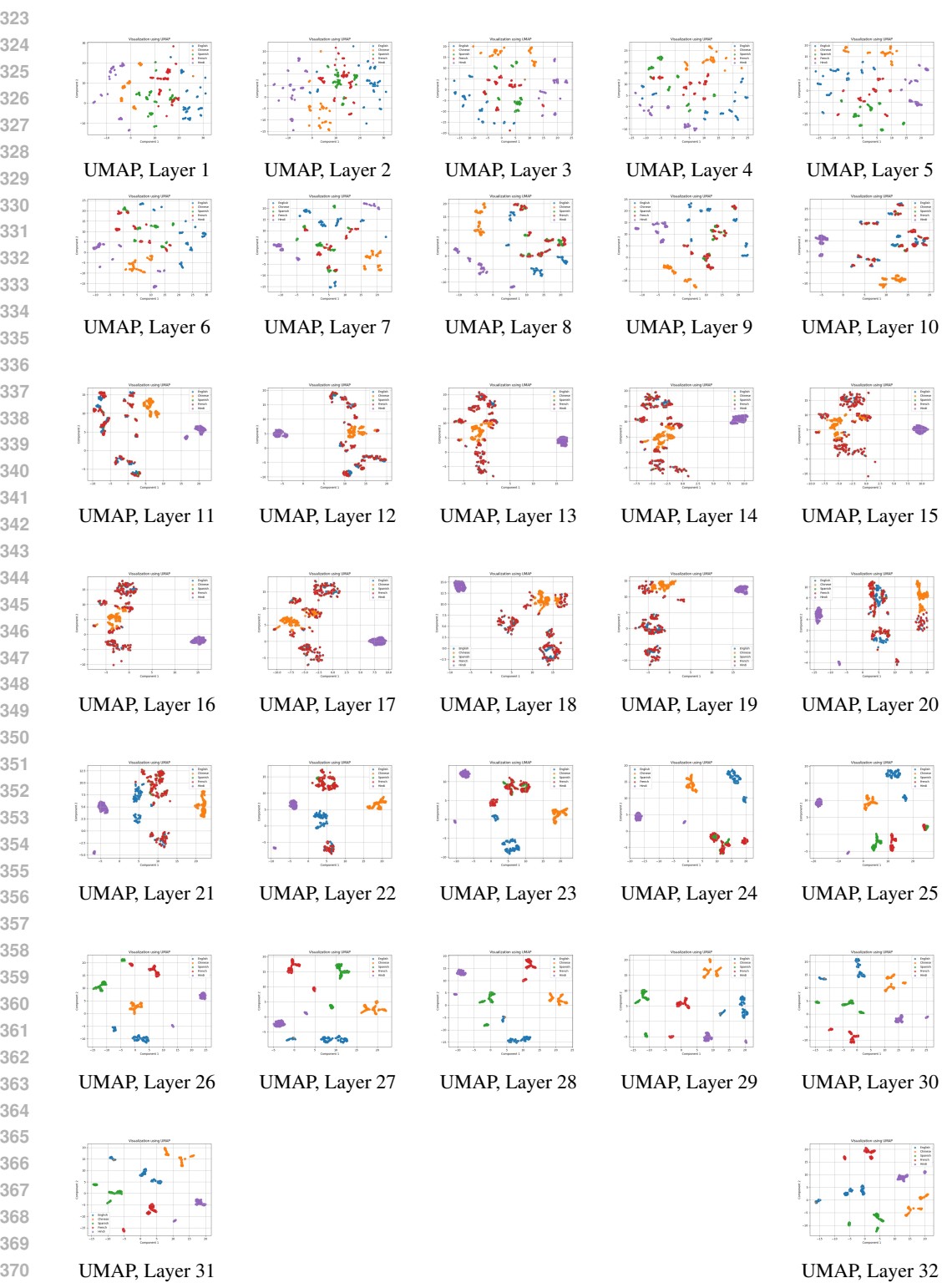

Figure 32: UMAP visualizations for layers 1-32 of Phi-3-mini-4k-instruct on the LogicalDeduction dataset.

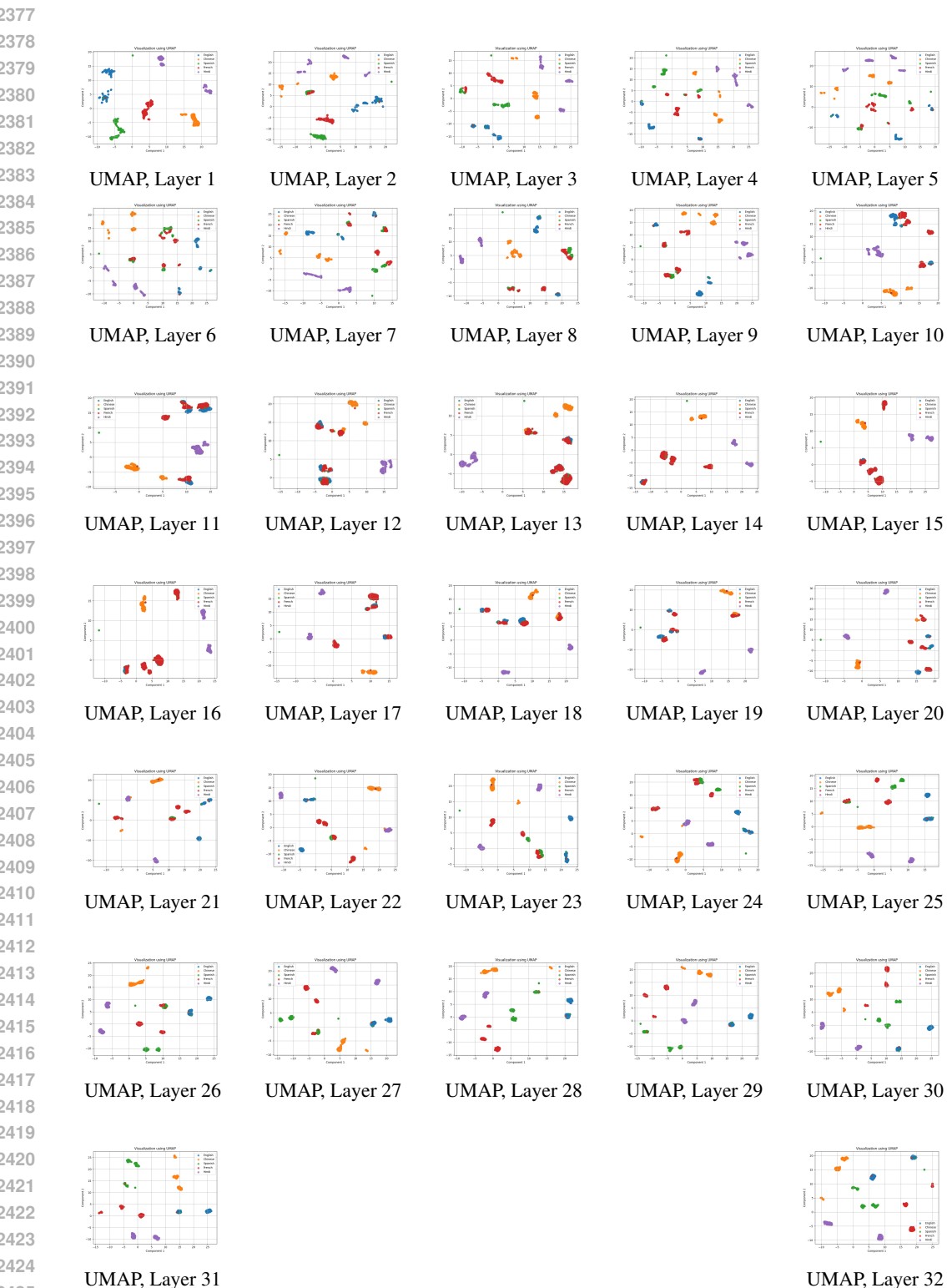

Figure 33: UMAP visualizations for layers 1-32 of Phi-3-mini-4k-instruct on the ProofWriter dataset.

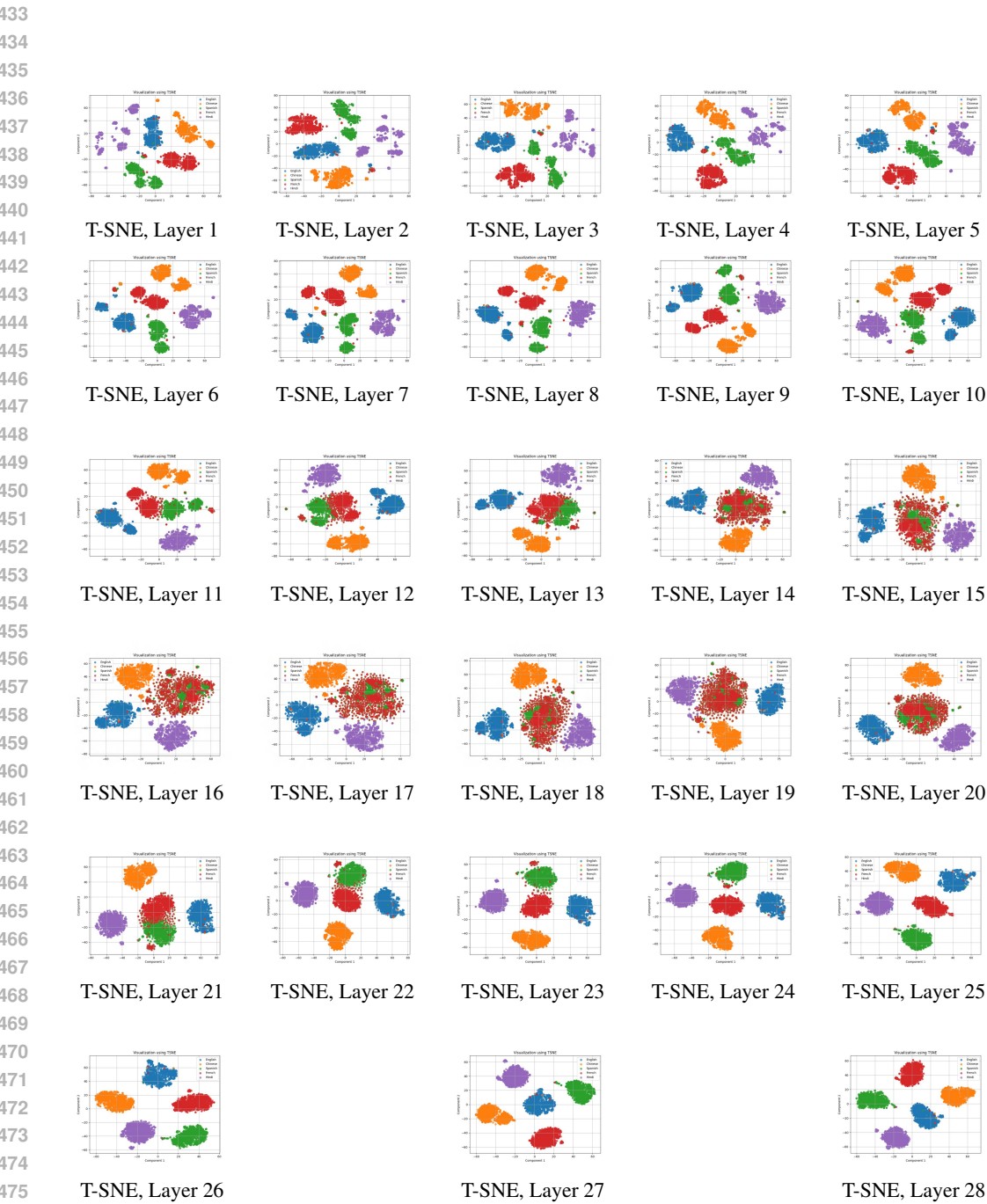

Figure 34: T-SNE visualizations for layers 1-28 of Qwen2-7B-Instruct on the GSM8K dataset.

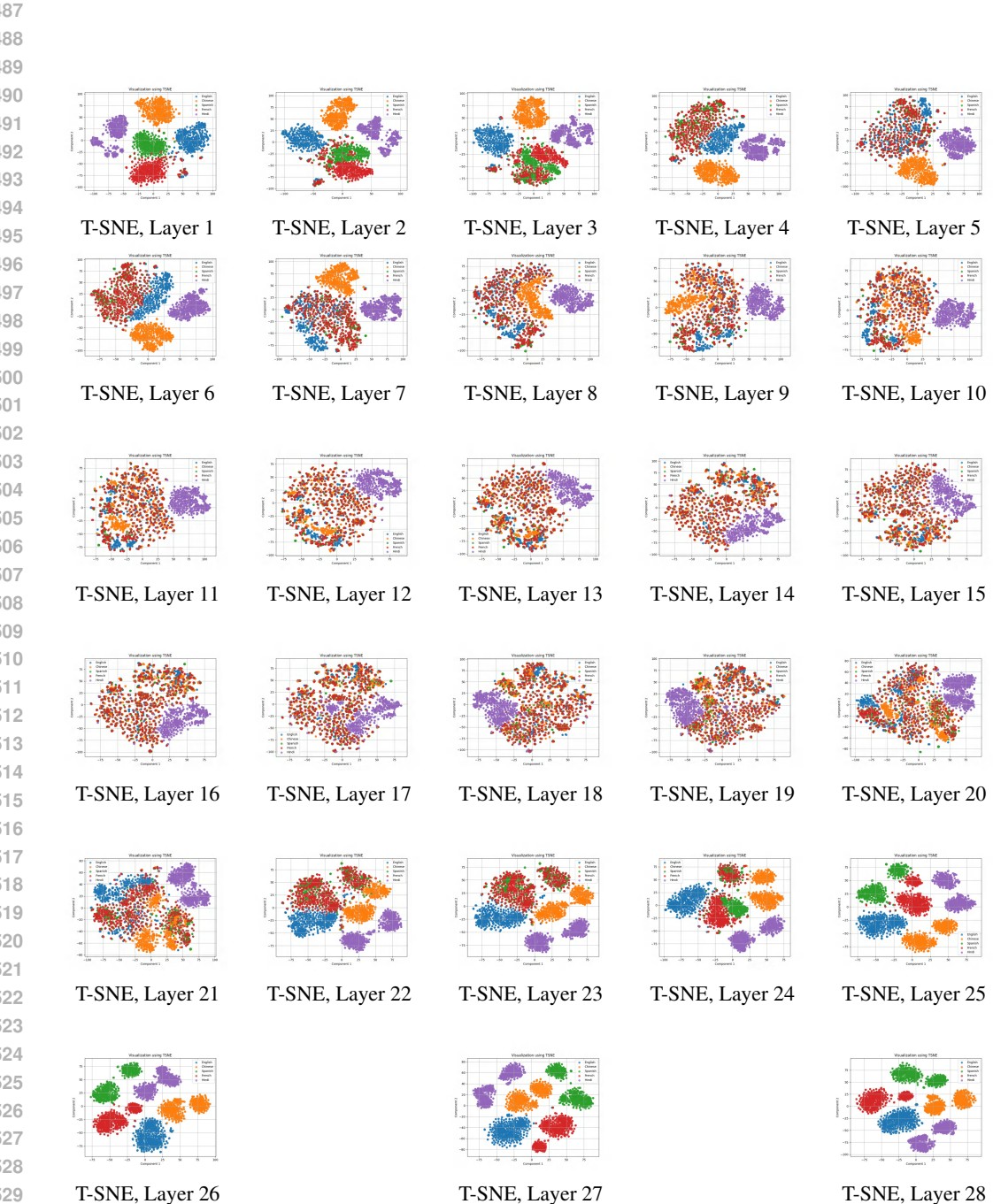

Figure 35: T-SNE visualizations for layers 1-28 of Qwen2-7B-Instruct on the FOLIO dataset.

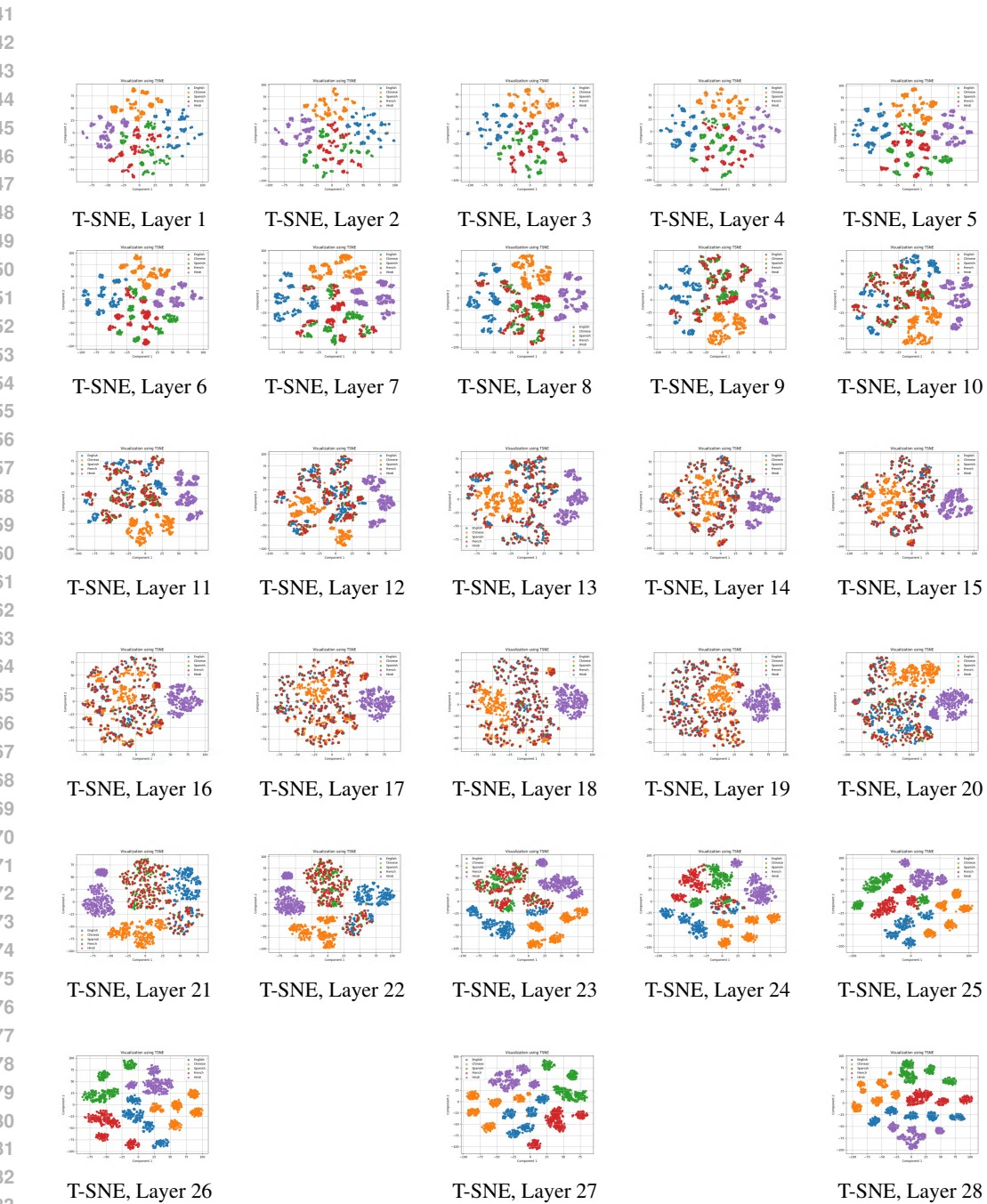

Figure 36: T-SNE visualizations for layers 1-28 of Qwen2-7B-Instruct on the LogicalDeduction dataset.

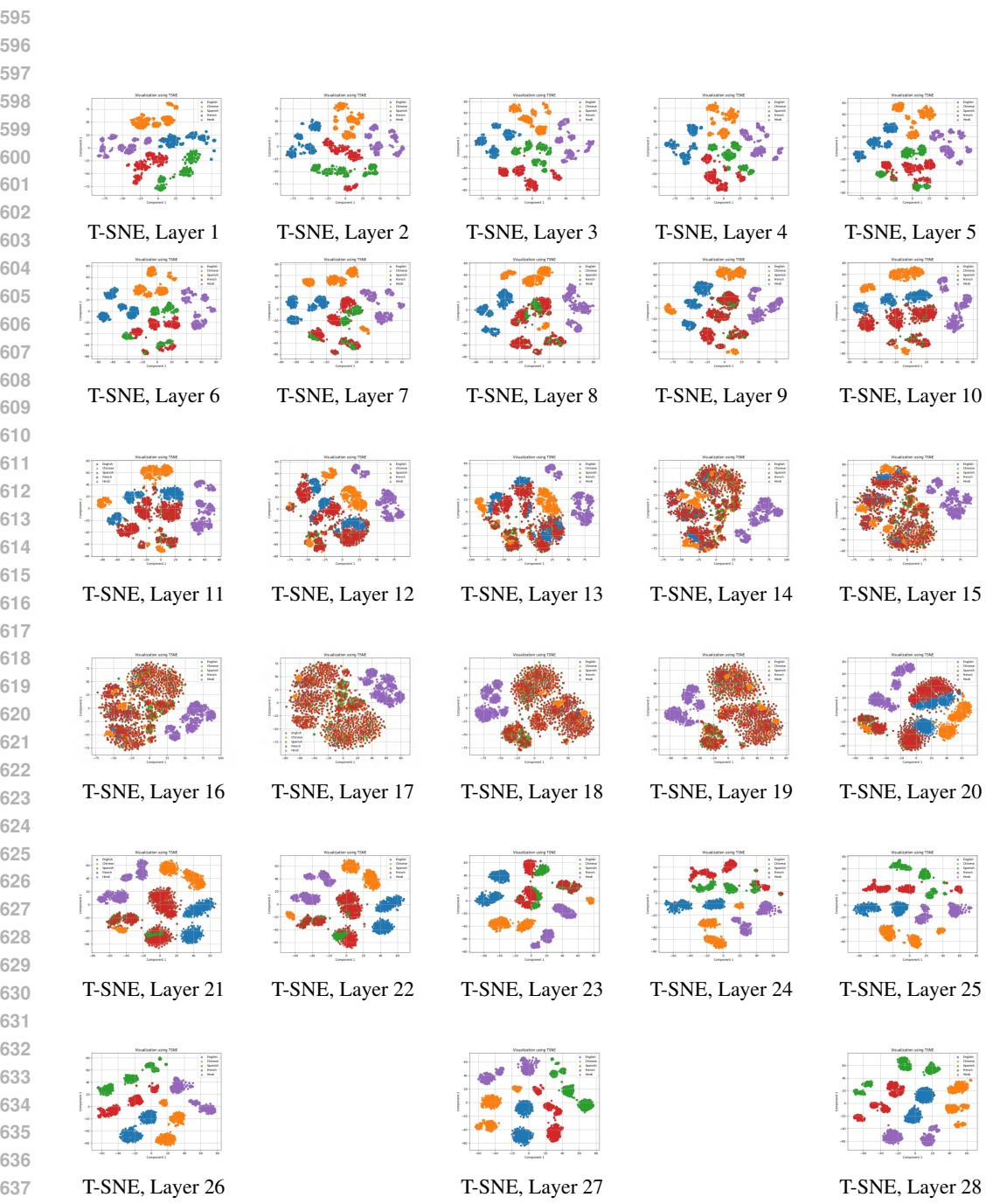

Figure 37: T-SNE visualizations for layers 1-28 of Qwen2-7B-Instruct on the ProofWriter dataset.

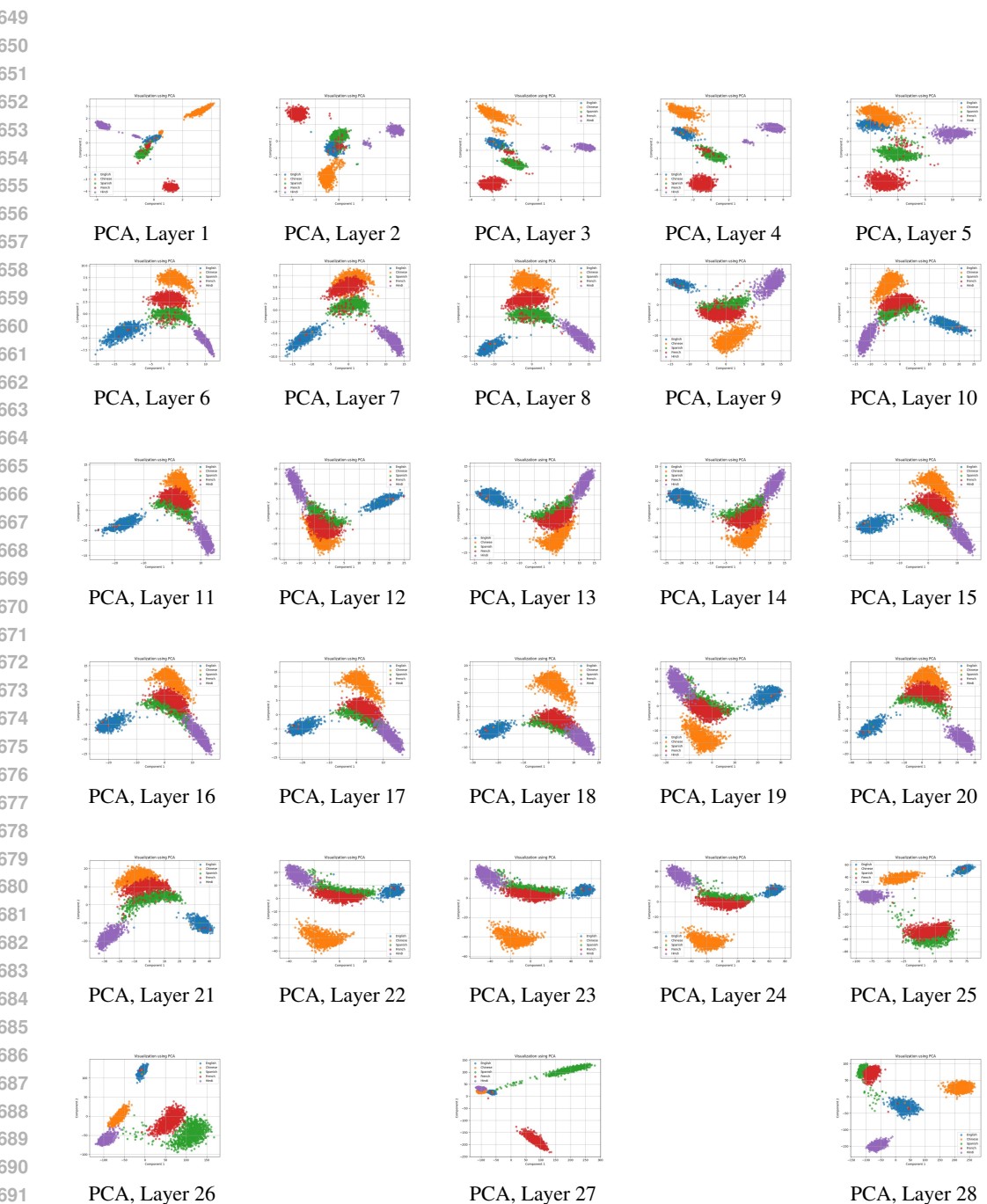

Figure 38: PCA visualizations for layers 1-28 of Qwen2-7B-Instruct on the GSM8K dataset.

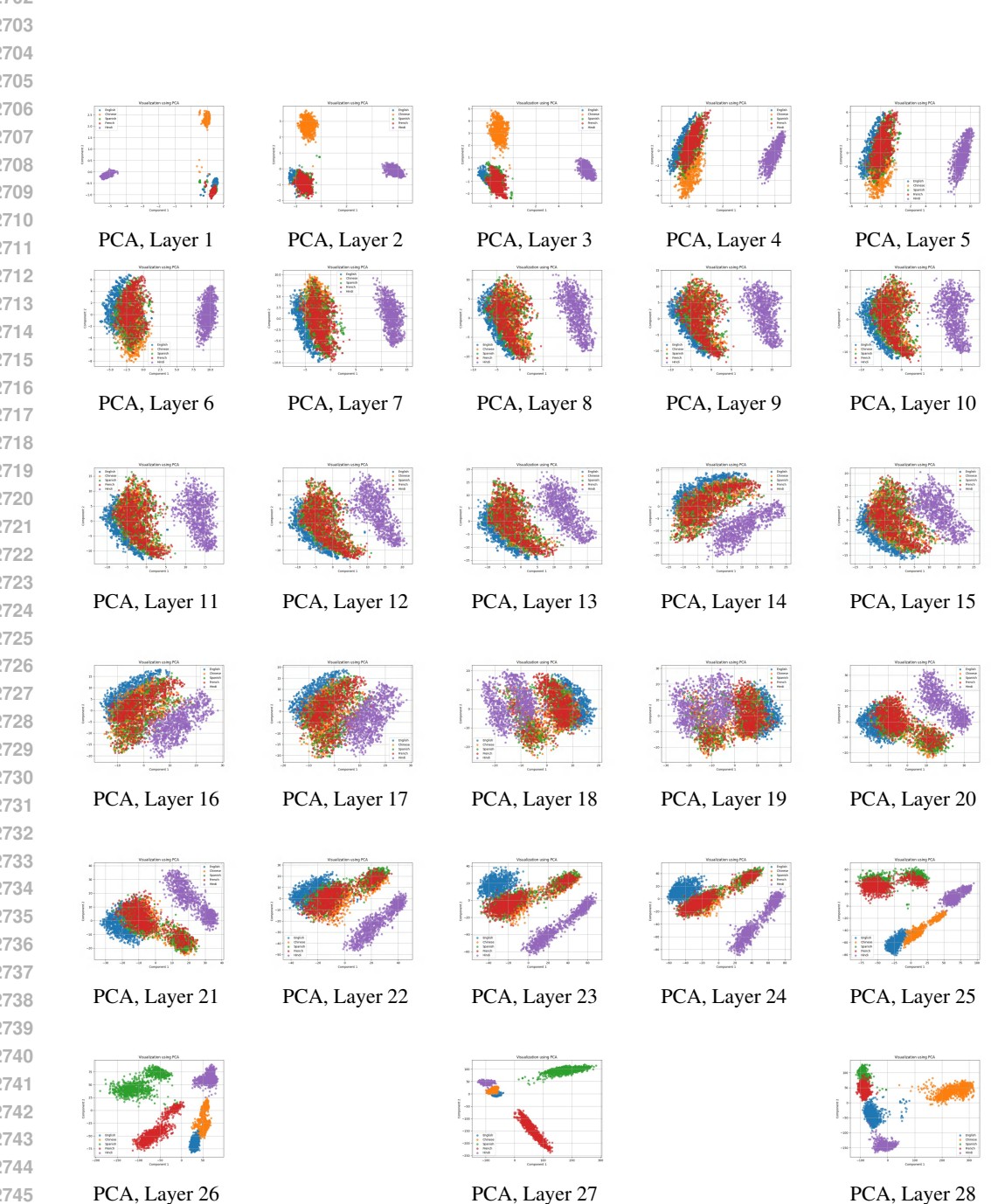

Figure 39: PCA visualizations for layers 1-28 of Qwen2-7B-Instruct on the FOLIO dataset.

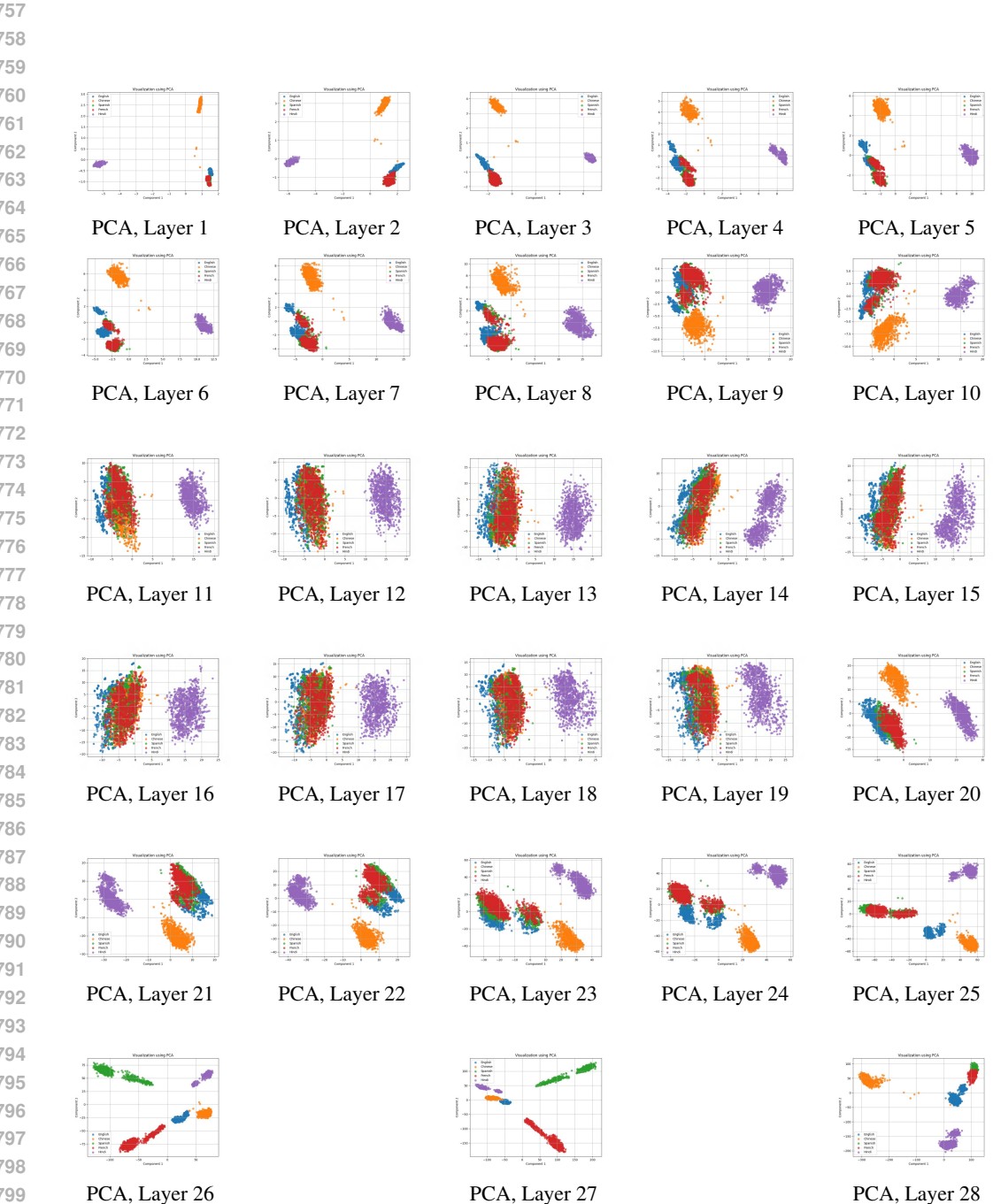

Figure 40: PCA visualizations for layers 1-28 of Qwen2-7B-Instruct on the LogicalDeduction dataset.

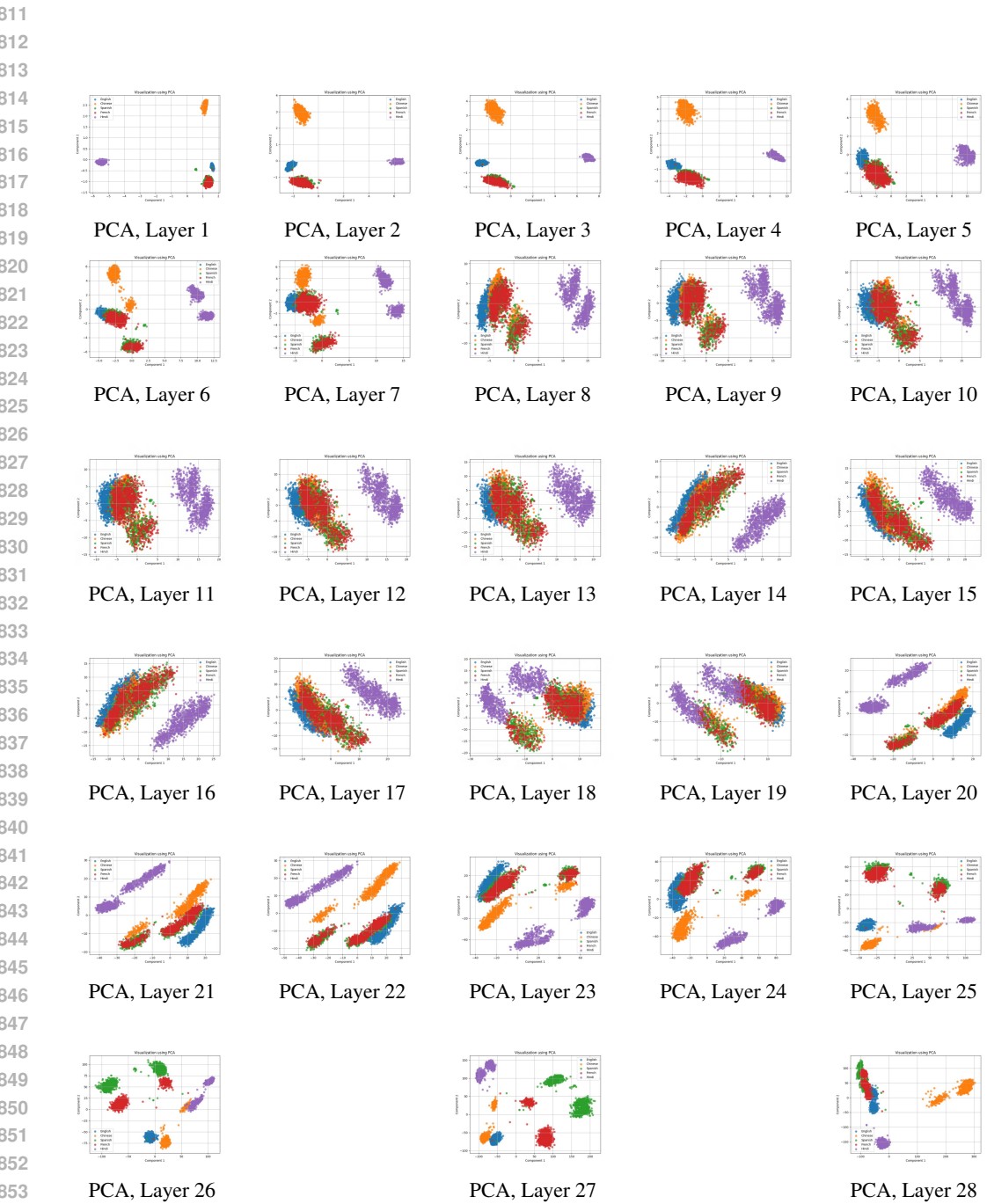

Figure 41: PCA visualizations for layers 1-28 of Qwen2-7B-Instruct on the ProofWriter dataset.

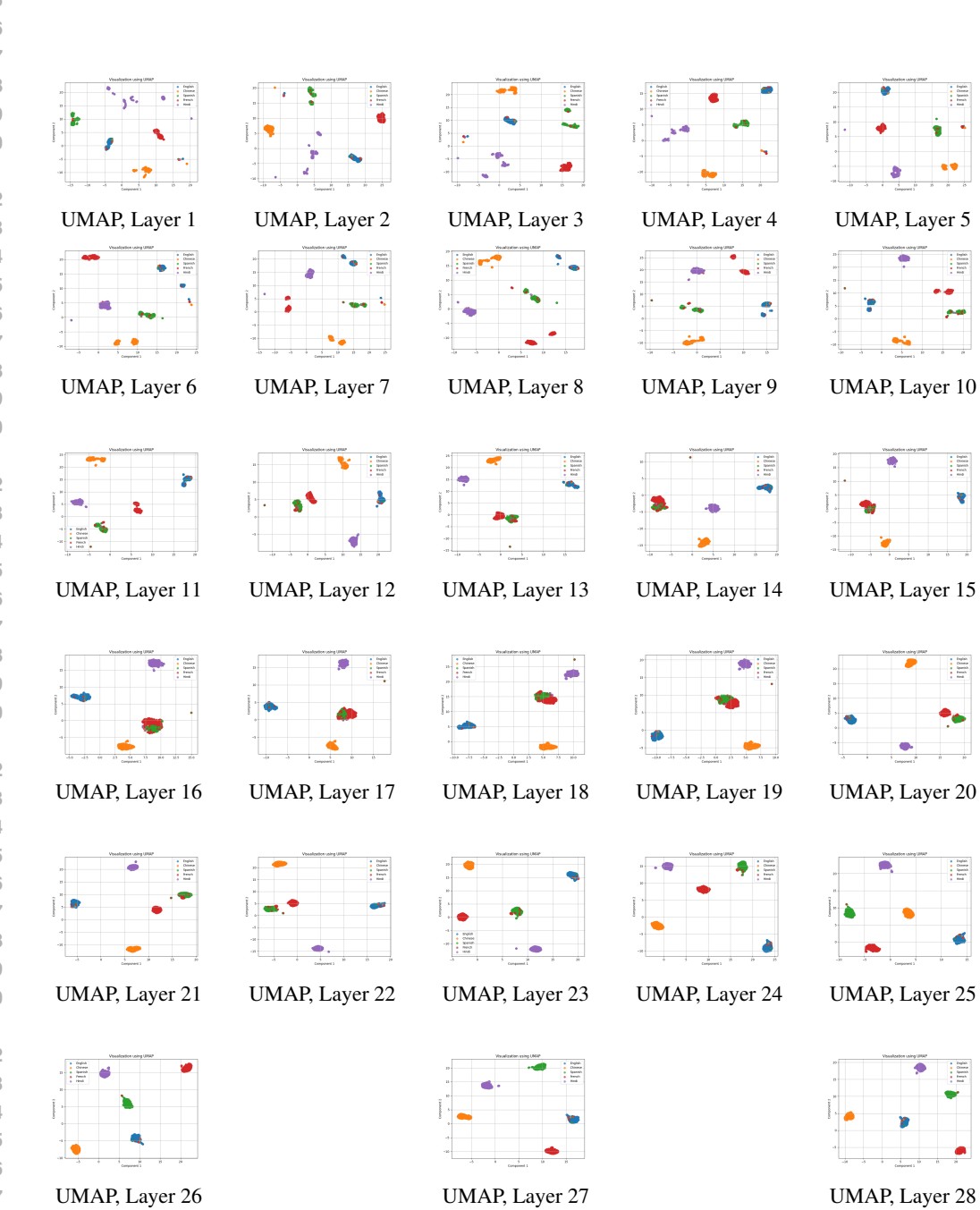

Figure 42: UMAP visualizations for layers 1-28 of Qwen2-7B-Instruct on the GSM8K dataset.

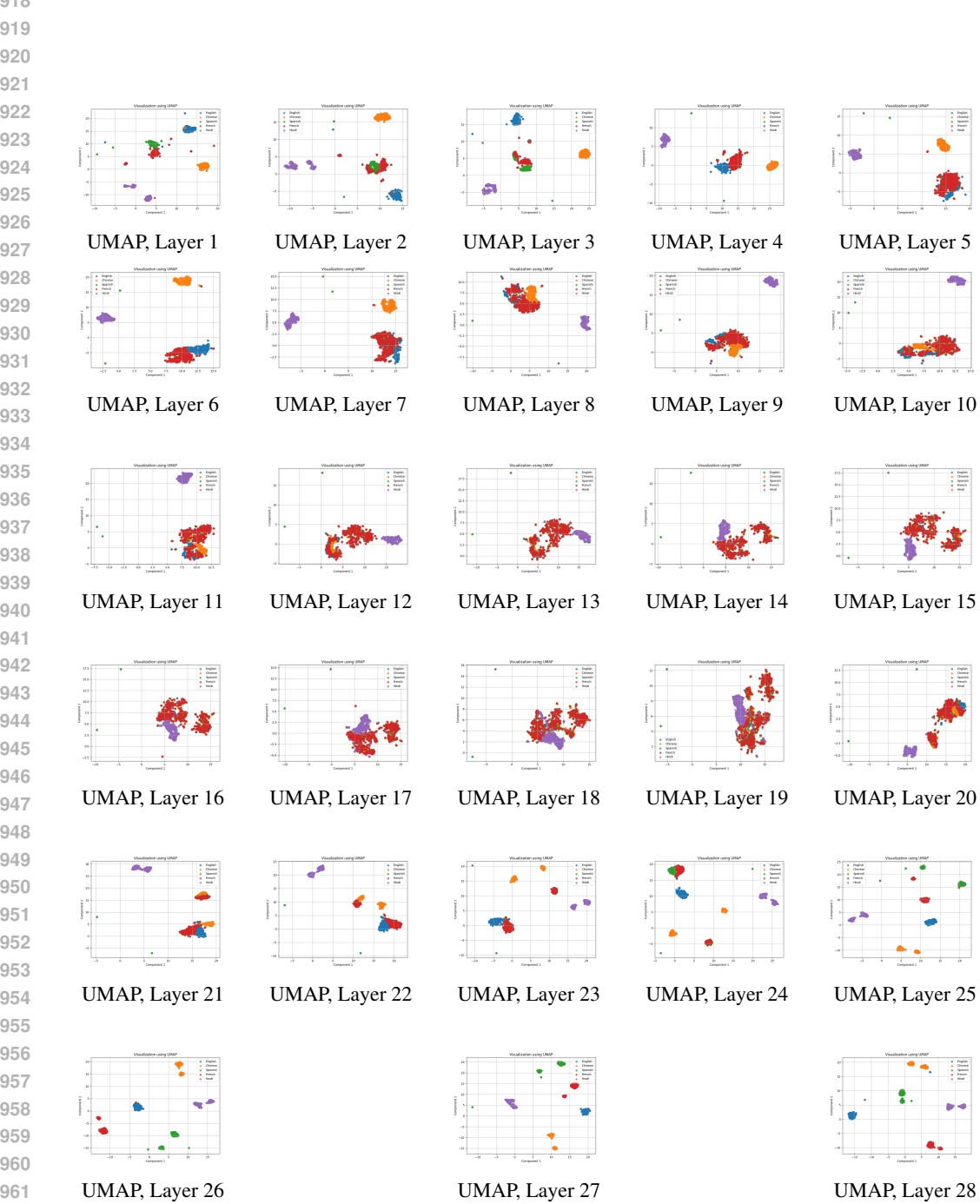

Figure 43: UMAP visualizations for layers 1-28 of Qwen2-7B-Instruct on the FOLIO dataset.

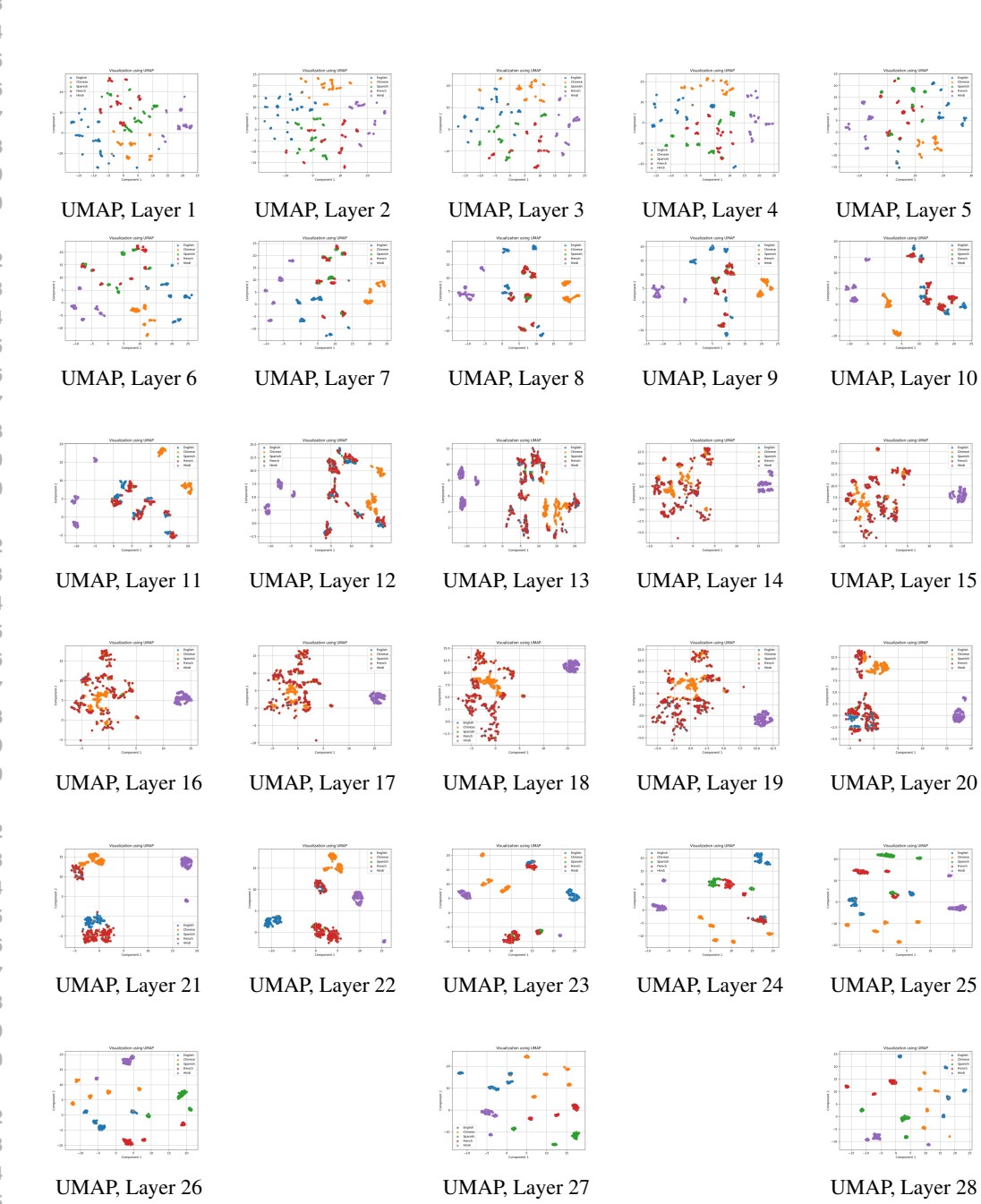

Figure 44: UMAP visualizations for layers 1-28 of Qwen2-7B-Instruct on the LogicalDeduction dataset.

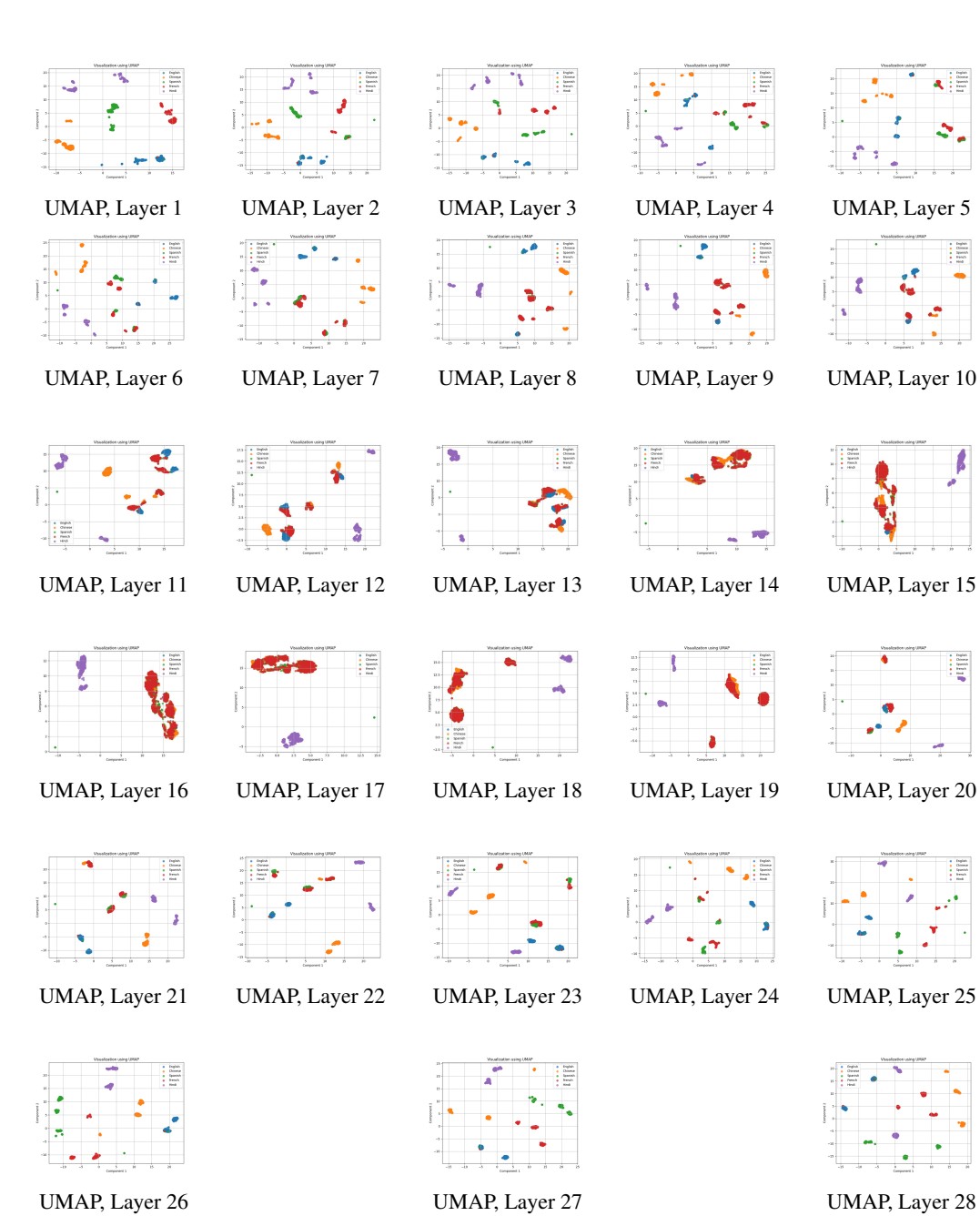

Figure 45: UMAP visualizations for layers 1-28 of Qwen2-7B-Instruct on the ProofWriter dataset.

## I  LLM USAGE

In the preparation of this manuscript, a large language model (LLM) was used to aid and polish the writing process. The authors have reviewed and edited all text and take full responsibility for the content of this paper.

