# OpenReview forum: "Layer-Informed Fine-Tuning via Three-Stage Functional Segmentation of LLMs"
_ICLR.cc/2026/Conference — ICLR 2026 Conference Withdrawn Submission_

### Official Review · Reviewer_hdv2 · 2025-10-23

**Soundness:** 3
**Presentation:** 2
**Contribution:** 3
**Rating:** 4
**Confidence:** 2

**Summary:**

This paper introduces and validates the "Three-Stage Functional Segmentation" hypothesis for Large Language Models (LLMs). The authors propose that an LLM's layers can be divided into three distinct functional stages: (1) conceptualization layers near the input, which transform natural language into abstract representations; (2) reasoning layers in the middle, which process these abstract concepts; and (3) textualization layers near the output, which convert the abstract results back into natural language. To validate this hypothesis, the authors conduct a series of experiments using cosine similarity of hidden states on multi-lingual and multi-task datasets. Based on the empirical evidence, they propose a method to identify the boundaries of these stages and further partition the reasoning stage into "active" and "idle" layers. Building on this framework, they introduce Layer-Informed Fine-Tuning (LIFT), a selective fine-tuning strategy that targets only the most task-relevant layers. Experimental results on several models and datasets show that LIFT can improve performance and reduce training time compared to standard full-model LoRA fine-tuning.

**Strengths:**

1. The three-stage functional segmentation framework is an intuitive way to conceptualize the inner workings of LLMs. It provides a clear, testable model that builds upon prior work in mechanistic interpretability but offers a more holistic view of the information flow.
2. The methodology used to test the hypothesis is well-designed. Using semantically equivalent questions in different languages (task-relevant) versus different questions in the same language (task-irrelevant) to track cosine similarity across layers provides strong and compelling evidence for the proposed segmentation. The resulting similarity curves are highly consistent and persuasive.
3. The proposed LIFT method is not just a theoretical finding but a practical technique for more efficient fine-tuning. The reported improvements in both performance and training time, however modest in some cases, are a valuable contribution to the field of parameter-efficient fine-tuning.

**Weaknesses:**

1. The methods for identifying the boundaries between stages, while functional, appear somewhat heuristic and lack rigorous justification.
- The boundary between conceptualization and reasoning  is determined by the first inflection point of the similarity curve. It is unclear if this is always the optimal or correct boundary, or how sensitive it is to the choice of dataset or languages.
- The boundary between active and idle reasoning relies on a hard-coded threshold (β = 0.94), which the authors state was chosen "empirically." This reduces the generality of the method. A sensitivity analysis on how this threshold affects layer segmentation and final task performance would significantly strengthen the paper.
2. The study is confined to relatively small-scale models (all under 10B parameters). As the authors acknowledge, it remains an open question whether this clean three-stage segmentation holds for much larger and deeper models (e.g., 70B models and some bigger). It is plausible that larger models exhibit more complex, overlapping, or distributed functional roles that cannot be so neatly segmented.
3. The experiments on the GSM8K dataset interestingly suggest that fine-tuning the conceptualization layers is more effective than tuning the reasoning layers for mathematical problems. Does this contradict that the middle layers are the universal "reasoning" hub. It suggests a task-dependent functionality that the paper could discuss in more detail. Does the optimal fine-tuning target depend on whether the task bottleneck is linguistic understanding versus abstract manipulation?
4. There is a noticeable inconsistency in the paper regarding the number of models used. The authors state in both the main body (sush as Section 3.1) and the conclusion that they employ four LLMs. However, the list provided in Section 3.1 only contains three models: Llama-3-8B-Instruct, Phi-3-mini-4k-instruct, and Qwen2-7B-Instruct. This typo should be corrected to ensure clarity and accuracy throughout the manuscript.

**Questions:**

1. Can you elaborate on the sensitivity of the boundary detection method? For instance, how does the choice of non-English language(s) in the cross-lingual similarity analysis affect the identified conceptualization-reasoning boundary?
2. Regarding the active-idle threshold β = 0.94: can you provide more details on how this specific value was determined? A graph showing performance vs. β value would be very insightful.
3. The finding that GSM8K benefits more from tuning conceptualization layers is fascinating. Do you hypothesize that this is because the core challenge for these smaller models is understanding the complex language of word problems, rather than the mathematical reasoning itself? Will this to change for models with stronger baseline mathematical abilities?

---

### Official Review · Reviewer_9m1p · 2025-10-25

**Soundness:** 2
**Presentation:** 4
**Contribution:** 2
**Rating:** 2
**Confidence:** 3

**Summary:**

This paper shows that LLM layers exhibit a structured division of labor across conceptualization, reasoning and textualization, through monitoring the inference process on cross-linguistic materials. In particular, they observe that LLMs transform natural language inputs into abstract representations, reason over these representations and transform back into natural language. They further propose a layer-specific fine-tuning approach.

**Strengths:**

- The paper is well-written so very easy to follow.
- The proposed layer-specific fine-tuning method outperforms full fine-tuning, with a lower cost.

**Weaknesses:**

1. The hypothesis is not surprising. Most, if not all, deep learning models transform raw input into some 'abstract' latent space to process and transform back to output space that is human-understandable.

2. I think the names of the different layers are misleading or subject to overclaiming, especially the reasoning layers (middle layers). Merely showing that the hidden states in these layers are similar on pairs of the same question in different languages is not enough to support that these layers are doing 'reasoning', it just shows that the same thing in different languages is represented in a similar way in the middle layers, arguably because of the multi-lingual pretraining.

3. The idea of fine-tuning a subset of layers is not new. The author could have discussed related works such as surgical fine-tuning [1] and other relevant works.

4. Experiments only compare the proposed method (LIFT with full fine-tuning, other representative PEFT baselines (i.e. selective fine-tuning methods that select a subset of weights for fine-tuning) is missing.

[1] Surgical fine-tuning improves adaptation to distribution shifts. Lee et al. 2023

**Questions:**

1. Which layers does LIFT fine-tune? For most datasets, reasoning layers are fine-tuned, but for GSM8K, conceptualization layers are fine-tuned. How do you decide which layers (conceptualization vs reasoning) to fine-tune for a given dataset?


2. The best performance seems always to be achieved without fine-tuning the textualization layers (i.e. layers close to the output side), can you explain the intuition behind? Intuitively, when fine-tuning conceptualization layers (input projection), the textualization layers (output projection) need to be adjust accordingly to match the shifts.


3. Can you clarify the process of your method when facing a new dataset? Do you need to translate the dataset into other languages first to identify funtionally-relavant layers through cosine similarity and then fine-tune? If yes, how would it work on datasets that are not language-intensive (like code data or math proof)?

Minor:
- The font of some figures (e.g. Figure 2) is too small to read.

---

### Official Review · Reviewer_erV7 · 2025-11-01

**Soundness:** 2
**Presentation:** 3
**Contribution:** 2
**Rating:** 4
**Confidence:** 3

**Summary:**

This paper explores how large language models (LLMs) perform complex reasoning and proposes a structured understanding of their internal inference process. The authors hypothesize that LLM layers follow a division of labour across three functional stages: conceptualization (transforming input text into abstract representations), reasoning(operating over these abstractions), and textualization (generating coherent natural language outputs). Building on this insight, they introduce LIFT, a finetuning strategy that selectively targets layers most relevant to a task’s functionality. Experiments demonstrate that LIFT enables more efficient finetuning and significant performance gains, offering new insights into both the interpretability and optimization of LLMs.

**Strengths:**

The novelty of this project lies in its layer-wise functional analysis of LLMs, identifying a structured division of labor across conceptualization, reasoning, and textualization stages. Based on this insight, the LIFT finetuning strategy selectively adapts only task-relevant layers, improving both efficiency and performance while providing a more interpretable approach to LLM optimization.

The research provides:

•	Enhanced interpretability by identifying a structured division of labor within LLM layers.

•	Efficient finetuning through selective adaptation of task-relevant layers, reducing training cost.

•	Improved performance demonstrated across tasks, showing practical benefits of the approach.

**Weaknesses:**

The following shortcomings could be addressed:

•	Enhanced interpretability by identifying a structured division of labor within LLM layers.

•	Efficient finetuning through selective adaptation of task-relevant layers, reducing training cost.

•	Improved performance demonstrated across tasks, showing practical benefits of the approach.

**Questions:**

In the abstract, you claim LIFT achieves efficient and effective fine-tuning. Can you describe which experiments demonstrate the efficiency of the method and which demonstrate the effectiveness of the fine-tuning?

While the experimental findings are presented as evidence supporting the stated hypothesis, the analysis does not include explicit testing of a null hypothesis. Without this, it is difficult to evaluate whether the observed effects could have occurred by chance. A more balanced hypothesis-testing framework would improve the credibility of the conclusions. Please discuss.

Can you describe which LLMs,  LIFT is suitable for? Is the approach that has been developed also suited to other technologies, such as LVLMs? How could you demonstrate the generalisability of LIFT to other transformer architectures?

---

### Note · Authors · 2025-11-16

I have read and agree with the venue's withdrawal policy on behalf of myself and my co-authors.